# Towards Generative Graph Matching for Graph Edit Distance Computation

Wei Huang [1]   Hanchen Wang [2]   Dong Wen [1]   Wenjie Zhang [1]   Ying Zhang [3]   Xuemin Lin [4]

## Abstract

Graph Edit Distance (GED), which aims to find an edit path with minimum number of edit operations to transform one graph into another, is a fundamental NP-hard problem and a widely used graph similarity measure. Recent matching-based hybrid approaches have demonstrated better scalability than A* search-based hybrids by reformulating GED as a graph matching problem. In these methods, a neural network predicts a single deterministic node matching matrix, from which top-$k$ node mappings are extracted iteratively to derive candidate edit paths. However, these methods often suffer from highly correlated candidates that easily lead to suboptimal solutions, while the iterative extraction becomes inefficient for large $k$. In this paper, we propose DiffGED, the first generative approach for GED computation. Specifically, we formulate the graph matching problem as a gnerative task, and employ a diffusion-based model to generate multiple diverse node matching matrices simultaneously, from which diverse node mappings can be efficiently extracted. The generative diversity introduced by the diffusion process enables DiffGED to avoid suboptimal solutions and achieve superior solution quality close to the exact solution. Experiments on real-world datasets show that DiffGED generates multiple diverse edit paths with accuracy comparable to exact solutions, while running faster than existing hybrid approaches. The source code is available at https://github.com/piupiupiuu/DiffGED.

## 1. Introduction

Graph Edit Distance (GED) is one of the most widely used similarity measures for graphs (Gouda & Arafa, 2015; Liang

[1]University of New South Wales, Australia [2]University of Technology Sydney, Australia [3]Zhejiang Gongshang University, China [4]Shanghai Jiaotong University, China. Correspondence to: Wei Huang <w.c.huang@unsw.edu.au>.

*Proceedings of the $43^{rd}$ International Conference on Machine Learning*, Seoul, South Korea. PMLR 306, 2026. Copyright 2026 by the author(s).

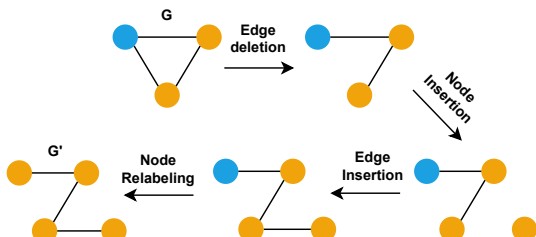

*Figure 1.* An optimal edit path for transforming $G$ to $G'$. $\mathrm{GED}(G, G') = 4$.

& Zhao, 2017; Bunke, 1997), with broad applications across computer vision and pattern recognition (Chen et al., 2020; Cho et al., 2013; Maergner et al., 2019). GED is defined as the minimum number of edit operations required to transform one graph into another. For instance, in Figure 1, transforming $G$ into $G'$ requires at least four edit operations, yielding $\mathrm{GED}(G, G') = 4$. However, due to its NP-hard nature, traditional A* search methods (Neuhaus et al., 2006; Blumenthal & Gamper, 2020; Chang et al., 2020) struggle to scale even to graphs with only a few nodes, as the search space grows exponentially with the number of nodes (Blumenthal & Gamper, 2020). In contrast, matching-based methods (Riesen & Bunke, 2009; Bunke et al., 2011) formulate GED computation as a bipartite graph matching problem and use linear-assignment-based heuristics to solve in polynomial time, but they often yield solutions of unsatisfactory quality.

In recent years, there has been growing interest in combining deep learning with traditional methods to compute GED more effectively and efficiently. Current state-of-the-art methods (Piao et al., 2023; Cheng et al., 2025) adopt a class of hybrid approaches that aim to enhance the solution quality of matching-based methods. Specifically, given a pair of graphs, a neural network (e.g., GNNs) is trained to predict a single node matching matrix, where a node matching matrix can be seen as a probability distribution. From this matrix, the top-$k$ discrete one-to-one node mappings with maximum matching weights are then extracted iteratively as shown in Figure 2, where each extracted mapping corresponds to a candidate edit path, and the candidate edit path with the minimum number of edit operations is selected as the final solution. However, this approach is deterministic (i.e., for

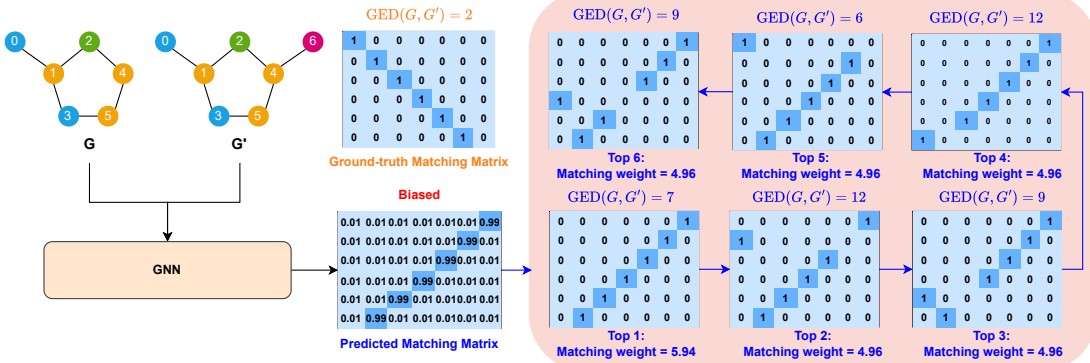

*Figure 2.* An example of existing matching-based hybrid approach that iteratively extracts top-$k$ maximum weight node mappings from a single deterministic node matching matrix predicted by GNN.

the same pair of graphs, it always produces the same deterministic output node matching matrix), and the extracted top-$k$ node mappings depend solely on a single predicted node matching matrix (i.e., probability distribution), with each mapping is extracted by searching on the previously extracted ones, leading to strong correlations among them. Thus, the following limitations could arise: (1) Highly correlated top-$k$ node mappings might easily fall into the local sub-optimal if the predicted node matching matrix is biased (i.e., significantly deviates from the correct matching). Consider the simple example of a biased predicted node matching matrix shown in Figure 2, it is clear to see that the top-6 node mappings extracted from the predicted matching matrix are highly correlated. Unfortunately, they are all sub-optimal with the derived GED significantly larger than the ground-truth GED; (2) Highly correlated node mappings limit the diversity of found edit paths, as multiple diverse edit paths could exist with multimodal distribution for an optimal GED; (3) The iterative extraction of top-$k$ node mappings is time consuming for large $k$, and cannot be parallelized to reduce the running time.

To address these limitations, we propose DiffGED, a novel generative approach that utilizes diffusion model for highly accurate GED computation. DiffGED first formulates matching-based GED computation as a conditional generation task, then it generates $k$ diverse and high-quality node matching matrices (i.e., probability distribution) in parallel from $k$ randomly initialized matrices, conditioned on the input graph pair, using our generative diffusion-based graph matching model DiffMatch. Next, $k$ candidate edit paths can be derived by extracting top-1 node mapping from each generated node matching matrix in parallel using a greedy algorithm. Therefore, comparing to previous deterministic approach, our proposed generative approach DiffGED offers the following advantages: (1) Each node mapping is extracted independently from a separate node matching matrix. With the stochasticity introduced by the generative diffusion model, the correlation between extracted node mappings is reduced, which enhances overall accuracy and decreases

the likelihood of the extracted candidate solutions being trapped in local optima; (2) The reduced correlation further improves the diversity of the discovered edit paths; (3) Both the $k$ node matching matrices and their corresponding node mappings can be generated and extracted in parallel, which greatly reduces runtime when $k$ is large.

**Contributions.** To the best of our knowledge, we are the first to introduce a generative formulation for solving graph matching and GED computation. We are also the first to leverage a generative diffusion model for graph matching, namely DiffMatch. Extensive experiments on real-world datasets demonstrate that our proposed DiffGED (1) has exceptionally high accuracy (around $95\%$ on all datasets) which outperforms the existing methods by a great margin, (2) can generate diverse edit paths, and (3) has a shorter running time compared to other hybrid approaches.

## 2. Related Work

**Traditional approaches.** Traditional approaches are often based on A* search (Neuhaus et al., 2006; Blumenthal & Gamper, 2020; Chang et al., 2020), guided by carefully designed heuristics to prune the unpromising search space. Unfortunately, these exact solvers are usually intractable for large graphs due to the NP-hard nature of GED computation. Traditional matching-based approaches improve scalability by constructing a node edition cost matrix, then model GED as a bipartite node matching problem and solve by either Hungarian (Riesen & Bunke, 2009) or VJ (Bunke et al., 2011) algorithm in polynomial time. However, they often yield poor solution quality.

**Regression-based deep learning approaches.** To address the limitations of traditional methods, deep learning approaches leverage the success of Graph Neural Networks (GNNs) in modeling complex graph structures. SimGNN (Bai et al., 2019) first formulated GED as a regression task with a cross-graph module, enabling fast and accurate prediction and inspiring many follow-ups (Bai & Zhao, 2021;

Zhuo & Tan, 2022; Ling et al., 2021; Bai et al., 2020; Zhang et al., 2021; Qin et al., 2021; Li et al., 2019; Jain et al., 2024). However, these methods are not trained to recover edit paths, which are crucial in many applications (Wang et al., 2021), and their predictions may underestimate GED without corresponding feasible edit paths.

**Hybrid approaches.** To recover the edit paths, hybrid approaches have been extensively studied, combining traditional search-based methods with deep learning techniques. A well-studied line of research focuses on guiding A* search with heuristics learned by a neural network (Yang & Zou, 2021; Wang et al., 2021; Liu et al., 2023), aiming to improve the efficiency of the search process. However, these methods often suffer from poor solution quality and inherit the scalability limitations of A* search. To improve both efficiency and effectiveness, recent state-of-the-art hybrid approaches such as GEDGNN (Piao et al., 2023) and GEDIOT (Cheng et al., 2025) have shifted towards improving the solution quality of matching-based approaches. These methods work by predicting a single node matching matrix via neural network, from which top-$k$ node mappings can be iteratively extracted to construct candidate edit paths. Compared with A* search-based hybrid approaches, this class of methods is significantly more efficient. However, they are still ineffective, since all candidate edit paths are derived from the same deterministic matching matrix, they exhibit high correlation and are prone to local optima. Moreover, the iterative node mapping extraction process is inherently sequential and cannot be parallelized, leading to inefficiency for large $k$. Taken together, these challenges suggest substantial room for improvement in hybrid GED computation.

## 3. Preliminaries

In this paper, we focus on the computation of graph edit distance between a pair of undirected labeled graphs $G = (V, E, L)$ and $G' = (V', E', L')$, where $G$ consists of a set of nodes $V$, a set of edges $E$ and a labeling function $L$ that assigns each node a label.

**Graph Edit Distance (GED).** Given a pair of graphs $(G, G')$, find an optimal edit path with minimum number of edit operations that transforms $G$ to $G'$. An edit path is a sequence of edit operations that transforms $G$ to $G'$. Graph edit distance $\text{GED}(G, G')$ is defined as the number of edit operations in the optimal edit path (Sanfeliu & Fu, 1983). Specifically, there are three types of edit operations: (1) insert or delete a node; (2) insert or delete an edge; (3) replace the label of a node.

**Edit path extraction.** Suppose $|V| \leq |V'|$, an edit path of transforming $G$ to $G'$ can be obtained from an injective node mapping $f$ from $V$ to $V'$ in linear time complexity $O(|V'|+|E|+|E'|)$ (Piao et al., 2023), such that $f(v) = v'$,

where $v \in V$ and $v' \in V'$. The overall procedure can be described as follows (the formal procedure can be found in Algorithm 1 of Appendix B.1):

(1) For each mapped node pair $f(v) = v'$, if $L(v) \neq L'(v')$, then replace the label of $v$ with $L'(v')$;

(2) For the remaining unmapped nodes in $V'$, insert $|V'| - |V|$ nodes into $V$. Each inserted node is mapped to and has the same label as an unmapped node in $V'$;

(3) For any two pairs of mapped nodes $f(v) = v'$ and $f(u) = u'$, if $(u, v) \in E$ and $(u', v') \notin E'$, delete the edge $(u, v)$ from $E$; if $(u, v) \notin E$ and $(u', v') \in E'$, insert the edge $(u, v)$ into $E$.

Therefore, to find an optimal edit path with minimum number of edit operations, we only have to find an optimal node mapping $f^*$.

## 4. Proposed Approach: DiffGED

### 4.1. DiffGED: Overview

As described in Section 3, the optimal edit path can be obtained from an optimal node mapping $f^*$. To approximately find the optimal node mapping $f^*$, one simple and effective way is to predict top-$k$ node mappings $f_1, ..., f_k$, then select the one that results in the minimum edit operations.

To obtain top-$k$ node mappings, our DiffGED adopts a two-phase approach, as shown in Figure 3. Specifically, in the first phase, given a graph pair $(G, G')$, instead of predicting a single node matching matrix (i.e., distribution), we predict top-$k$ node matching matrices $\hat{M}_1, ..., \hat{M}_k$ simultaneously, where each element in $\hat{M}_i \in \mathbb{R}^{|V| \times |V'|}$ represents the matching weight of a pair of nodes. Then, in the second phase, a simple greedy algorithm is used to extract top-1 node mapping independently from each predicted node matching matrix $\hat{M}_i$ in parallel, such that $f_i = Top1(\hat{M}_i)$. Comparing to existing matching-based approaches, our approach yields the following benefits: (1) Phase 1 reduces the correlation between each node mapping extracted in Phase 2, thus decreases the chances of falling into sub-optimal; (2) The reduced correlation naturally improves the diversity of the extracted node mappings; (3) Both the prediction of node matching matrices (Phase 1) and the extraction of node mappings (Phase 2) can be fully parallelized, significantly reducing the overall running time.

However, the neural networks in existing matching-based approaches cannot be easily adapted to Phase 1 of our approach. This is because they are deterministic and have limited capacity to predict a flexible number of node matching matrices for a given input graph pair. Once trained, they can only produce a fixed number of node matching matrices (often just one), and this requires a corresponding num-

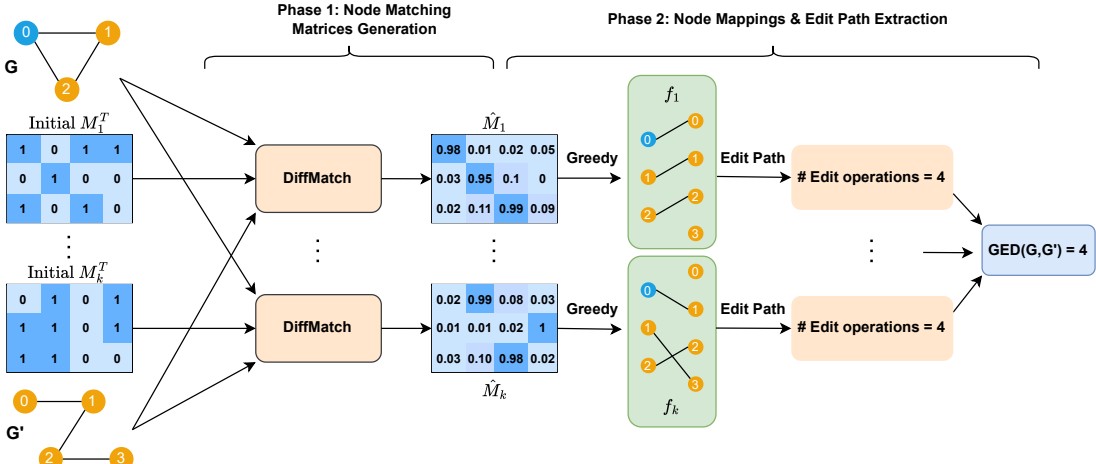

*Figure 3.* An overview of DiffGED. In the first phase, DiffGED first samples $k$ random initial node matching matrices, then DiffMatch will denoise the sampled node matching matrices via diffusion model. In the second phase, one node mapping will be extracted from each node matching matrix in parallel, and edit paths will be derived from the node mappings.

ber of prediction heads in the network architecture, which consequently increases the number of unnecessary network parameters. Even worse, the node matching matrices predicted by different heads often remain highly correlated, as they share the same inputs and deterministic backbone, which inherently lack stochasticity.

To predict a flexible number of diverse node matching matrices, a generative approach can be naturally well-suited to this improved two-phase approach. For Phase 1, we propose DiffMatch, a generative graph matching model that generates $k$ diverse and high-quality node matching matrices in parallel. As shown in Figure 3, unlike deterministic models that rely solely on the graph pair as input, our generative model DiffMatch introduces stochasticity by taking a randomly initialized discrete node matching matrix $M_i^T \in \{0,1\}^{|V| \times |V'|}$ as an additional input. It then denoises the sampled $M_i^T$, conditioned on the graph pair, to generate $\hat{M}_i \in \mathbb{R}^{|V| \times |V'|}$. This design enables the flexible generation of $k$ distinct node matching matrices in parallel by sampling $k$ random initial node matching matrices $M_1^T, ..., M_k^T$, with $k$ chosen arbitrarily at inference time and independent of the training phase. Therefore, it eliminates the need for multiple prediction heads. Moreover, this generative formulation is also motivated by the fact that multiple optimal node mappings could exist with multimodal distribution for a given graph pair. Thus, different initial random node matching matrices can be mapped to different optima to reduce the correlation among the generated node matching matrices.

To further enhance the generation of high-quality and diverse node matching matrices, our DiffMatch leverages the generative diffusion model (Ho et al., 2020; Dhariwal & Nichol, 2021; Sohl-Dickstein et al., 2015; Song & Ermon, 2019) to denoise each $M_i^T$, which has demonstrated im-

pressive success in image generation tasks, but has not yet been explored in the context of graph matching. The main strength of diffusion model over other generative models is that it enables the generation of node matching matrices through an iterative refinement process, breaking down the complex generation task into simpler steps. Each step makes minor adjustments, progressively improving the quality of the matching matrices. Furthermore, each refinement step introduces stochasticity, which further reduces the correlation between generated node matching matrices and enhances the model's ability to produce diverse node matching matrices. To handle discrete data, we adopt discrete diffusion (Haefeli et al., 2022; Vignac et al., 2022; Austin et al., 2021) for DiffMatch.

### 4.2. Phase 1: DiffMatch

In this section, we introduce our DiffMatch based on a single discrete node matching matrix $M \in \{0,1\}^{|V| \times |V'|}$.

**DiffMatch overview.** Diffusion models are generative models that consist of a forward process and a reverse process. Given a ground-truth node matching matrix $M^0$ (transformed from the ground-truth node mapping), the forward process $q(M^{1:T}|M^0) = \prod_{t=1}^{T} q(M^t|M^{t-1})$ of DiffMatch progressively corrupts $M^0$ to a sequence of increasingly noisy latent variables $M^{1:T} = M^1, M^2, ..., M^T$. Then, the reverse process of DiffMatch learns to reconstruct $M^{t-1}$ from $M^t$ using a denoising network. During inference (Figure 4), the learned reverse process progressively denoises the latent variables towards the desired distribution, starting from a randomly sampled noise $M^T$, such that: $p_\theta(M^{0:T}|G, G') = p(M^T) \prod_{t=1}^{T} p_\theta(M^{t-1}|M^t, G, G')$.

**Forward process & Training.** The training procedure of DiffMatch can be found in Algorithm 2 of Appendix B.2.

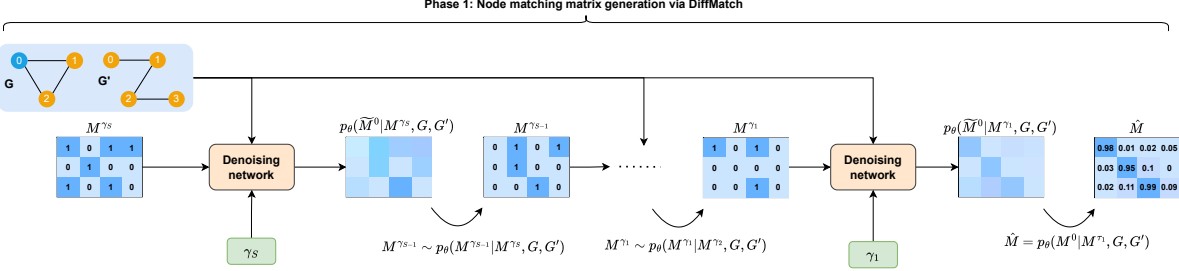

*Figure 4.* Reverse process of diffusion-based node matching model DiffMatch during inference.

Let $\widetilde{M}^t \in \{0,1\}^{|V| \times |V'| \times 2}$ be the one-hot encoding of the node matching matrix $M^t$ at time step $t \in [0,T]$. The forward process adds noise to $M^{t-1}$ and samples $M^t$ from the following Categorical distribution: $q(M^t|M^{t-1}) = \text{Cat}(M^t|p = \widetilde{M}^{t-1}Q_t)$, with the transition probability matrix $Q_t = \begin{bmatrix} 1 - \beta_t & \beta_t \\ \beta_t & 1 - \beta_t \end{bmatrix}$, where $\beta_t$ is the probability of switching node matching state.

In practice, to sample the noisy matching matrix $M^t$ efficiently during training, we can compute the $t$-step marginal from $M^0$, such that: $q(M^t|M^0) = \text{Cat}(M^t|p = \widetilde{M}^0\overline{Q}_t)$, with $\overline{Q}_t = Q_1Q_2...Q_t$. Then, the denoising network is trained to predict node matching probabilities $p_\theta(\widetilde{M}^0|M^t, G, G')$ that reconstructs $M^0$ from $M^t$ by minimizing the binary cross-entropy loss:

$$\mathcal{L} = \frac{1}{|V||V'|} \sum_{v \in V} \sum_{v' \in V'} (M^0[v][v'] \log(p_\theta(\widetilde{M}^0|M^t, G, G')[v][v']))$$
$$+ (1 - M^0[v][v']) \log(1 - p_\theta(\widetilde{M}^0|M^t, G, G')[v][v'])) \quad (1)$$

**Reverse process & Inference.** The overall reverse process of DiffMatch is presented in Figure 4 (the algorithm can be found in Algorithm 3 of Appendix B.3). With the trained denoising network, each step $t$ of the reverse process can then denoise $M^t$ to $M^{t-1}$ as:

$$M^{t-1} \sim p_\theta(M^{t-1}|M^t, G, G')$$
$$= \sum_{\widetilde{M}} q(M^{t-1}|M^t, \widetilde{M}^0) p_\theta(\widetilde{M}^0|M^t, G, G') \quad (2)$$

$$q(M^{t-1}|M^t, M^0) = \frac{q(M^t|M^{t-1}, M^0)q(M^{t-1}|M^0)}{q(M^t|M^0)}$$
$$= \text{Cat}(M^{t-1}; p = \frac{\bar{M}^t Q_t^\top \odot \bar{M}^0 \overline{Q}_{t-1}}{\bar{M}^0 \overline{Q}_t (\bar{M}^t)^\top}) \quad (3)$$

where $q(M^{t-1}|M^t, M^0)$ denotes the posterior, with $\bar{M} \in \{0,1\}^{|V||V'| \times 2}$ obtained by reshaping $\widetilde{M} \in \{0,1\}^{|V| \times |V'| \times 2}$. During inference, starting from a random noisy discrete node matching matrix $M^T$, each $M_{t-1}$ can be sampled from $p_\theta(M^{t-1}|M^t, G, G')$ via Bernoulli sampling. And note that, for the last reverse step (i.e., $t-1 = 0$), we directly use $\hat{M} = p_\theta(M^{t-1}|M^t, G, G')$ as the input of the node mapping extraction in Phase 2.

Notably, during training, the forward process typically employs a large number of steps $T$ (e.g., $T = 1000$), and performing $T$ reverse steps during inference can be computationally expensive. To further accelerate DiffMatch's inference, we apply DDIM (Song et al., 2020) to the reverse process. The key idea of DDIM is that, instead of performing $T$ reverse steps over the entire sequence $[T, ..., 1]$, we perform only $S$ reverse steps on a sub-sequence $[\tau_S, ..., \tau_1]$ of $[T, ..., 1]$, where $S < T$ and $\tau_S = T$. We substitute $t$ and $t - 1$ in Equation 2 with $\tau_i$ and $\tau_{i-1}$, and we rewrite the posterior as follows:

$$q(M^{\tau_{i-1}}|M^{\tau_i}, M^0) = \frac{q(M^{\tau_i}|M^{\tau_{i-1}}, M^0)q(M^{\tau_{i-1}}|M^0)}{q(M^{\tau_i}|M^0)}$$
$$= \text{Cat}(M^{\tau_{i-1}}; p = \frac{\bar{M}^{\tau_i} \overline{Q}_{\tau_{i-1}, \tau_i}^\top \odot \bar{M}^0 \overline{Q}_{\tau_{i-1}}}{\bar{M}^0 \overline{Q}_{\tau_i} (\bar{M}^{\tau_i})^\top})$$
$$(4)$$

where $\overline{Q}_{\tau_{i-1}, \tau_i} = Q_{\tau_{i-1}+1}Q_{\tau_{i-1}+2}...Q_{\tau_i}$.

**Matching-aware denoising network.** Notably, the network architectures used in existing deterministic matching-based approaches can only take graph pairs as inputs and thus cannot be applied to our generative formulation to denoise the noisy matching matrix. Therefore, we design a matching-aware denoising network as shown in Figure 5. The network takes as input the graph pair $(G, G')$, the noisy node matching matrix $M^t$ along with its transpose $M^{t\top}$, and the corresponding time step $t$. Intuitively, it then works by directly computing the embeddings of each node matching pair, and predicting the node matching probabilities $p_\theta(\widetilde{M}^0|M^t, G, G')$ based on these embeddings to reconstruct $M^0$. Note that, $\text{GED}(G, G') = \text{GED}(G', G)$, we assume symmetry in node matching: if node $v \in V$ matches with node $v' \in V'$, then $v'$ also matches with $v$. Therefore, we only sample $M^t \in \mathbb{R}^{|V| \times |V'|}$ during both training and inference, then use both $M^t$ and $M^{t\top}$ as inputs to the denoising network.

For more details, let $\boldsymbol{h}_v^l$ and $\boldsymbol{h}_{v'}^l$ denote the embedding of node $v \in V$ and $v' \in V'$ at layer $l$, $\boldsymbol{h}_{vv'}^l$ and $\boldsymbol{h}_{v'v}^l$ denote the embedding of node matching pair $(v, v')$ and $(v', v)$ at layer $l$. For initialization, the node embeddings $\boldsymbol{h}_v^0$ and $\boldsymbol{h}_{v'}^0$ are initialized as the one-hot node labels, the node matching pair embeddings $\boldsymbol{h}_{vv'}^0$ and $\boldsymbol{h}_{v'v}^0$ are initialized as the sinu-

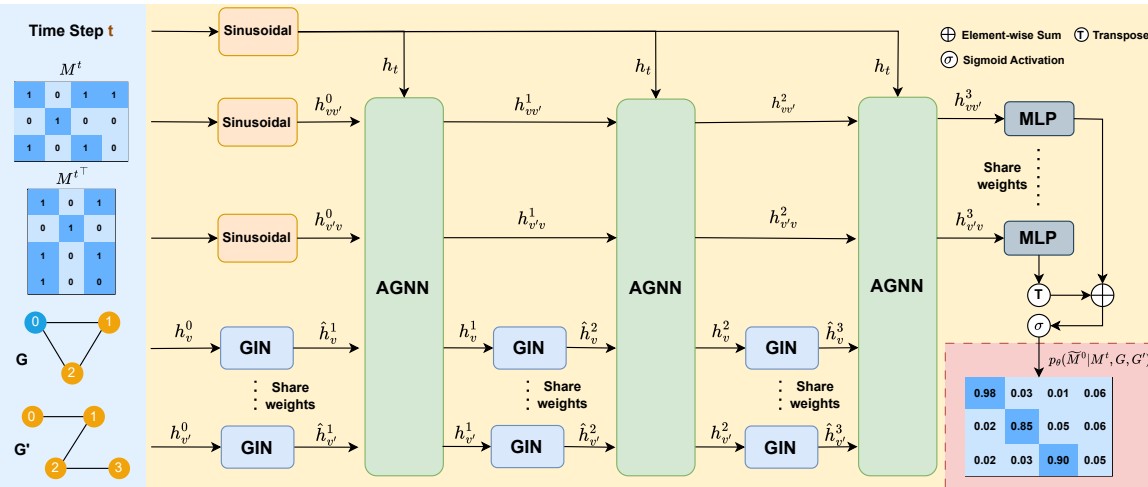

*Figure 5.* An overview of the matching-aware denoising network. The blue area denotes the network input, the yellow area denotes the architecture of the denoising network, and the pink area denotes the network output.

soidal embeddings (Vaswani et al., 2017) of corresponding values in $M^t$ and $M^{t\top}$, and the time step embedding $\boldsymbol{h}_t$ is initialized as the sinusodial embedding of $t$.

For each layer $l$, the denoising network first updates the node embeddings of each graph to $\hat{\boldsymbol{h}}_v^l$ and $\hat{\boldsymbol{h}}_{v'}^l$, independently using their respective graph structures (intra-graph) via GIN (Xu et al., 2018). Then, the denoising network further refines the embeddings to $\boldsymbol{h}_v^l$ and $\boldsymbol{h}_{v'}^l$, while also updating the node matching pair embeddings to $\boldsymbol{h}_{vv'}^l$ and $\boldsymbol{h}_{v'v}^l$, by incorporating noisy interactions between node matching pairs (inter-graph) and the time step $t$ through Anisotropic Graph Neural Network (AGNN) (Joshi et al., 2020; Sun & Yang, 2023; Qiu et al., 2022). The key advantage of AGNN is its ability to directly update embeddings for node matching pairs, enabling more expressive representations for cross-graph tasks. In contrast, traditional GNNs such as GIN are specifically designed for computing node embeddings only, which makes them less suitable for handling noisy node matching matrices in our generative formulation. More details about AGNN can be found in Appendix B.4.

Finally, the matching-aware denoising network computes the matching values of each node pair via multi-layer perceptron (MLP), and sums the matching values for corresponding pairs $(v, v')$ and $(v', v)$, then applies sigmoid activation to obtain the node matching probabilities $p_\theta(\widetilde{M}^0|M^t, G, G')$.

### 4.3. Phase 2: Node Mapping Extraction

After sampling $k$ noisy node matching matrices $M_1^T, ..., M_k^T$ and denoising to $\hat{M}_1, ..., \hat{M}_k$, we adopt the greedy algorithm based on matching weights to extract one node mapping from each node matching matrix (the detailed procedure can be found in Algorithm 4 of Appendix B.5.). Specifically, assuming $|V| \leq |V'|$, the

greedy node mapping extraction starts by selecting the node pair with the highest matching probability. Once a node pair is selected, all matching probabilities involving either of the selected nodes are set to $-\infty$ to prevent them from being selected again. This process is repeated iteratively $|V|$ times until every node in $V$ is assigned to a corresponding node in $V'$.

Note that, the above greedy algorithm does not guarantee the extraction of optimal node mappings from the node matching matrices, but it has a time complexity of $O(|V|^2|V'|)$ slightly faster than the exact Hungarian algorithm (Kuhn, 1955) with time complexity of $O(|V'|^3)$. It can also be easily parallelized by GPU to extract $k$ node mappings from $k$ node matching matrices simultaneously to reduce the running time, especially for large $k$. It will be demonstrated in Appendix D.3 that DiffGED with the above greedy algorithm is sufficient to achieve excellent performance.

## 5. Experiments

### 5.1. Experimental Settings

**Datasets.** We conduct experiments over three popular real-world GED datasets: AIDS700 (Bai et al., 2019), Linux (Wang et al., 2012; Bai et al., 2019) and IMDB (Bai et al., 2019; Yanardag & Vishwanathan, 2015). For each dataset, we split 60%, 20%, and 20% of all the graphs as training set, validation set, and testing set, respectively. To form training/validation/testing graph pairs, as well as their corresponding ground-truth labels, we follow the same strategy described in (Piao et al., 2023). Due to space limitations, more details about datasets can be found in Appendix C.1.

**Evaluation metrics.** We evaluate our DiffGED against baseline methods using the feasible GED derived from the predicted node mappings based on the following metrics: (1)

*Table 1.* Overall performance on cross-train-test and intra-test graph pairs. Methods with a running time exceeding 24 hours are marked with -.

| Datasets | Settings | Models | MAE ↓ | Accuracy ↑ | $\rho$ ↑ | $\tau$ ↑ | p@10 ↑ | p@20 ↑ | Time(s) ↓ |
|---|---|---|---|---|---|---|---|---|---|
| AIDS700 | Cross-train-test | Hungarian | 8.247 | 1.1% | 0.547 | 0.431 | 52.8% | 59.9% | 0.00011 |
| | | VJ | 14.085 | 0.6% | 0.372 | 0.284 | 41.9% | 52% | 0.00017 |
| | | Noah | 3.057 | 6.6% | 0.751 | 0.629 | 74.1% | 76.9% | 0.6158 |
| | | GENN-A* | 0.632 | 61.5% | 0.903 | 0.815 | 85.6% | 88% | 2.98919 |
| | | GEDGNN | 1.098 | 52.5% | 0.845 | 0.752 | 89.1% | 88.3% | 0.39448 |
| | | GEDIOT | 1.188 | 53.5% | 0.825 | 0.73 | 88.6% | 86.7% | 0.39357 |
| | | MATA* | 0.838 | 58.7% | 0.8 | 0.718 | 73.6% | 77.6% | **0.00487** |
| | | DiffGED (ours) | **0.022** | **98%** | **0.996** | **0.992** | **99.8%** | **99.7%** | 0.0763 |
| | Intra-test | Hungarian | 8.237 | 1.5% | 0.527 | 0.416 | 54.3% | 60.3% | 0.0001 |
| | | VJ | 14.171 | 0.9% | 0.391 | 0.302 | 44.9% | 52.9% | 0.00016 |
| | | Noah | 3.174 | 6.8% | 0.735 | 0.617 | 77.8% | 76.4% | 0.5765 |
| | | GENN-A* | 0.508 | 67.1% | 0.917 | 0.836 | 87.1% | 90.6% | 3.44326 |
| | | GEDGNN | 1.155 | 50.5% | 0.838 | 0.746 | 89.1% | 87.6% | 0.39344 |
| | | GEDIOT | 1.348 | 47.4% | 0.81 | 0.71 | 88.4% | 86.9% | 0.39707 |
| | | MATA* | 0.885 | 56.6% | 0.77 | 0.689 | 73.2% | 76.6% | **0.00486** |
| | | DiffGED (ours) | **0.024** | **96.4%** | **0.993** | **0.986** | **99.7%** | **99.7%** | 0.07546 |
| Linux | Cross-train-test | Hungarian | 5.35 | 7.4% | 0.696 | 0.605 | 74.8% | 79.6% | 0.00009 |
| | | VJ | 11.123 | 0.4% | 0.594 | 0.5 | 72.8% | 76% | 0.00013 |
| | | Noah | 1.596 | 9% | 0.9 | 0.834 | 92.6% | 96% | 0.24457 |
| | | GENN-A* | 0.213 | 89.4% | 0.954 | 0.905 | 99.1% | 98.1% | 0.68176 |
| | | GEDGNN | 0.094 | 96.6% | 0.979 | 0.969 | 98.9% | 99.3% | 0.12863 |
| | | GEDIOT | 0.117 | 95.3% | 0.978 | 0.966 | 98.8% | 99% | 0.13535 |
| | | MATA* | 0.18 | 92.3% | 0.937 | 0.893 | 88.5% | 91.8% | **0.00464** |
| | | DiffGED (ours) | **0.0** | **100%** | **1.0** | **1.0** | **100%** | **100%** | 0.06982 |
| | Intra-test | Hungarian | 5.423 | 7.5% | 0.725 | 0.623 | 75% | 77% | 0.00008 |
| | | VJ | 11.174 | 0.4% | 0.613 | 0.512 | 70.6% | 74.5% | 0.00013 |
| | | Noah | 1.879 | 8% | 0.872 | 0.796 | 84.3% | 92.2% | 0.25712 |
| | | GENN-A* | 0.142 | 92.9% | 0.976 | 0.94 | 99.6% | 99.6% | 1.17702 |
| | | GEDGNN | 0.105 | 96.2% | 0.979 | 0.968 | 98.6% | 98.5% | 0.12169 |
| | | GEDIOT | 0.14 | 94.8% | 0.973 | 0.959 | 98.1% | 98.3% | 0.12826 |
| | | MATA* | 0.201 | 91.5% | 0.948 | 0.903 | 86.2% | 90.2% | **0.00464** |
| | | DiffGED (ours) | **0.0** | **100%** | **1.0** | **1.0** | **100%** | **100%** | 0.06901 |
| IMDB | Cross-train-test | Hungarian | 21.673 | 45.1% | 0.778 | 0.716 | 83.8% | 81.9% | 0.0001 |
| | | VJ | 44.078 | 26.5% | 0.4 | 0.359 | 60.1% | 62% | 0.00038 |
| | | Noah | - | - | - | - | - | - | - |
| | | GENN-A* | - | - | - | - | - | - | - |
| | | GEDGNN | 2.469 | 85.5% | 0.898 | 0.879 | 92.4% | 92.1% | 0.42428 |
| | | GEDIOT | 2.822 | 84.5% | 0.9 | 0.878 | 92.3% | 92.7% | 0.41959 |
| | | MATA* | - | - | - | - | - | - | - |
| | | DiffGED (ours) | **0.937** | **94.6%** | **0.982** | **0.973** | **97.5%** | **98.3%** | **0.15105** |
| | Intra-test | Hungarian | 21.156 | 45.9% | 0.776 | 0.717 | 84.2% | 82.1% | 0.00012 |
| | | VJ | 44.072 | 26.6% | 0.4 | 0.359 | 60.1% | 63.1% | 0.00037 |
| | | Noah | - | - | - | - | - | - | - |
| | | GENN-A* | - | - | - | - | - | - | - |
| | | GEDGNN | 2.484 | 85.5% | 0.895 | 0.876 | 92.3% | 91.7% | 0.42662 |
| | | GEDIOT | 2.83 | 84.4% | 0.989 | 0.876 | 92.5% | 92.4% | 0.42269 |
| | | MATA* | - | - | - | - | - | - | - |
| | | DiffGED (ours) | **0.932** | **94.6%** | **0.982** | **0.974** | **97.5%** | **98.4%** | **0.15107** |

*Mean Absolute Error (MAE)* measures the average absolute difference between the predicted GED and the ground-truth GED; (2) *Accuracy* measures the ratio of the testing graph pairs with predicted GED equals to the ground-truth GED; (3) *Spearman's Rank Correlation Coefficient ($\rho$)*, and (4) *Kendall's Rank Correlation Coefficient ($\tau$)*, both measure the matching ratio between the ranking results of graphs based on their predicted GEDs and the ground-truth GEDs for each query testing graph; (5) *Precision at top-$10/20$ (p@10/20)* measure the ratio of predicted top-$10/20$ similar graphs within the ground-truth top-$10/20$ similar graphs for each query testing graph; (6) *Time(s)* measures the average running time over all testing graph pairs.

**Baselines & Implementation details.** Details of baseline methods can be found in Appendix C.2. The implementation details can be found in Appendix C.3.

## 5.2. Main Results

**Overall performance on cross-train-test graph pairs.** Table 1 presents the overall performance of all methods on the standard cross-train-test pairs (i.e., each pair is formed by a testing graph with a training graph) evaluation setting. Across all datasets, DiffGED demonstrates exceptionally high solution quality in terms of MAE, accuracy, and all ranking metrics. For the AIDS700 dataset, the accuracy of DiffGED is nearly double that of other hybrid approaches. DiffGED consistently shows shorter running times than most hybrid approaches across all datasets, although it is slower than MATA* on smaller datasets. Note that, all A*-based hybrid approaches fail to complete evaluations (on IMDB) within a reasonable time due to the scalability issues inherent in A* search.

Specifically, both MATA* and DiffGED need to predict

*Table 2.* Overall performance on cross-train-test pairs and intra-test graph pairs without structural train-test leakage.

| Datasets | Settings | Models | MAE ↓ | Accuracy ↑ | $\rho$ ↑ | $\tau$ ↑ | p@10 ↑ | p@20 ↑ | Time(s) ↓ |
|---|---|---|---|---|---|---|---|---|---|
| AIDS700 | Cross-train-test | GEDGNN | 1.148 | 51.8% | 0.836 | 0.742 | 88.9% | 88.2% | 0.39227 |
| | | GEDIOT | 1.159 | 53.8% | 0.832 | 0.737 | 89.6% | 89% | 0.39381 |
| | | DiffGED (ours) | **0.046** | **96%** | **0.992** | **0.983** | **99.8%** | **99.6%** | **0.07431** |
| | Intra-test | GEDGNN | 1.235 | 48.7% | 0.824 | 0.729 | 90.1% | 88.4% | 0.39079 |
| | | GEDIOT | 1.349 | 46.5% | 0.8 | 0.701 | 88.1% | 86.9% | 0.39263 |
| | | DiffGED (ours) | **0.064** | **94.4%** | **0.987** | **0.975** | **99.5%** | **99.5%** | **0.07438** |
| Linux | Cross-train-test | GEDGNN | 0.935 | 65.1% | 0.809 | 0.722 | 85.8% | 87.4% | 0.27418 |
| | | GEDIOT | 1.009 | 65.3% | 0.788 | 0.706 | 87.9% | 85% | 0.27556 |
| | | DiffGED (ours) | **0.165** | **92.4%** | **0.958** | **0.931** | **93.7%** | **95.3%** | **0.07277** |
| | Intra-test | GEDGNN | 1.335 | 57.6% | 0.755 | 0.664 | 85.3% | 100% | 0.28935 |
| | | GEDIOT | 1.435 | 52.6% | 0.772 | 0.686 | 86.3% | 100% | 0.29775 |
| | | DiffGED (ours) | **0.305** | **86.1%** | **0.896** | **0.857** | **92.1%** | **100%** | **0.07694** |
| IMDB | Cross-train-test | GEDGNN | 4.799 | 73.1% | 0.817 | 0.783 | 85.2% | 85.5% | 0.75584 |
| | | GEDIOT | 4.679 | 74.9% | 0.826 | 0.794 | 87.6% | 86.8% | 0.73493 |
| | | DiffGED (ours) | **1.12** | **94%** | **0.973** | **0.963** | **97.1%** | **97.1%** | **0.22247** |
| | Intra-test | GEDGNN | 4.822 | 73.1% | 0.822 | 0.789 | 85.9% | 86.1% | 0.75577 |
| | | GEDIOT | 4.689 | 74.8% | 0.829 | 0.797 | 87.9% | 87% | 0.74122 |
| | | DiffGED (ours) | **1.141** | **93.8%** | **0.971** | **0.961** | **97%** | **97.2%** | **0.22315** |

node matching matrices and then extract top-$k$ candidate results. However, they differ in key aspects: (1) MATA* predicts only two node matching matrices in a single step, whereas DiffGED generates $k$ node matching matrices in parallel over 10 denoising steps. This results in faster node matching matrix prediction for MATA*; (2) MATA* extracts the top-$k$ candidate matching nodes in $G'$ for each node in $G$, limiting the valid range of $k$ to $|V'|$ and typically selecting a small $k$ to reduce the A* search space. In contrast, DiffGED extracts the top-$k$ global maximum weight node mappings, allowing $k$ to be arbitrarily large. As a result, MATA* achieves shorter running times on smaller datasets. However, on larger datasets, MATA* suffers from the exponential growth of the A* search space, whereas DiffGED remains unaffected by this limitation.

Moreover, while GEDGNN and GEDIOT can scale to large graphs, they are both slower and perform worse across all datasets for several reasons. GEDGNN and GEDIOT iteratively extract top-$k$ candidate node mappings from a single predicted node matching matrix, resulting in highly correlated mappings. In contrast, DiffGED extracts top-$k$ candidate node mappings from $k$ different node matching matrices in parallel, generating diverse mappings. This diversity reduces the likelihood of falling into local sub-optimal solutions, even if some generated node matching matrices are biased. Additionally, the parallelization of node mapping extraction significantly reduces runtime.

**Generalization on unseen intra-test graph pairs.** To evaluate the generalization ability of DiffGED on unseen graphs, we additionally consider a more challenging setting beyond the standard cross-train-test pairs commonly used in previous works. Specifically, we construct intra-test pairs, where each pair consists of two unseen testing graphs. Table 1 presents the overall performance of all methods on these unseen intra-test graph pairs. Compared to the results of

cross-train-test pairs in Table 1, it demonstrates that DiffGED can still achieve superior performance without losing solution quality.

**Generalization on datasets without structural train–test leakage.** Notably, AIDS, Linux, and IMDB datasets have recently been shown to suffer from structural train–test leakage (Roy et al., 2025), meaning that there exist isomorphic graphs in these datasets. This leakage may cause the reported results to overestimate the true generalization ability of each method. To address this concern, we follow the procedure described in (Roy et al., 2025) to remove all isomorphic graphs to obtain unique graphs, and then form training and testing pairs using only these unique graphs. Table 2 reports the performance of each method on both cross train-test pairs and intra-test graph pairs after removing structural train-test leakage. Compared with the results in Table 1, the performance of DiffGED decreases slightly in relative terms (e.g., the MAE on the AIDS cross train-test pairs approximately doubles) after removing train-test leakage. Nevertheless, DiffGED still consistently outperforms the baseline methods and achieves near-optimal performance across all datasets. Moreover, in terms of absolute performance degradation, the baseline methods exhibit substantially larger drops than DiffGED on both Linux and IMDB. These results further demonstrate the strong generalization ability of DiffGED.

### 5.3. Generative Top-$k$ vs. Iterative Top-$k$

To better evaluate the effectiveness, efficiency, and edit-path diversity of our generative top-$k$ approach, which extracts $k$ diverse node mappings from $k$ matching matrices, we compare it with the iterative approach commonly used in existing matching-based frameworks (e.g., GEDGNN, GEDIOT), which extracts highly correlated node mappings from a single node matching matrix (i.e., distribution). Specifically,

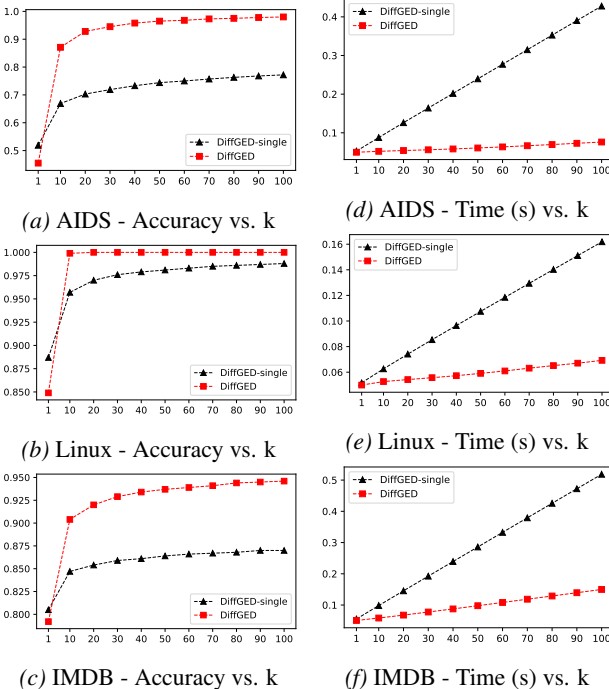

*(a)* AIDS - Accuracy vs. k  *(d)* AIDS - Time (s) vs. k

*(b)* Linux - Accuracy vs. k  *(e)* Linux - Time (s) vs. k

*(c)* IMDB - Accuracy vs. k  *(f)* IMDB - Time (s) vs. k

*Figure 6.* Effectiveness and efficiency of Top-$k$ approaches on cross-train-test pairs.

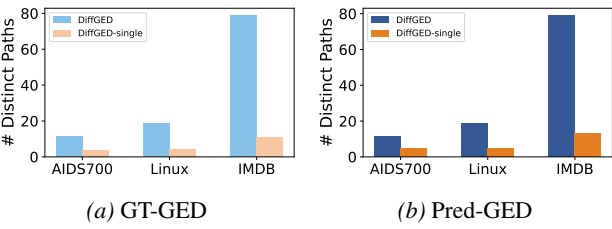

*(a)* GT-GED  *(b)* Pred-GED

*Figure 7.* Evaluation of Found Edit Path Diversity on cross-train-test pairs. Pred-GED refers to average number of distinct edit paths with predicted minimum GED. GT-GED refers to average number of distinct edit paths with ground-truth GED.

we create a variant model, DiffGED-single, which generates only one node matching matrix using DiffMatch and then applies the iterative top-$k$ extraction.

**Effectiveness.** As shown in Figure 6(a)-(c), our top-$k$ approach (DiffGED) performs slightly worse than DiffGED-single when $k = 1$. This is because DiffGED-single obtains the top-1 node mapping using the exact Hungarian algorithm, whereas DiffGED derives the top-1 mapping from the same node matching matrix via an approximate greedy algorithm. However, as $k$ increases, this initial disadvantage diminishes, with DiffGED rapidly converging to near-optimal accuracy, even with its approximate greedy algorithm. In contrast, DiffGED-single, despite using an exact extraction algorithm, converges to sub-optimal accuracy.

Notably, for simpler datasets like Linux, DiffGED achieves optimal solution quality with a small value of $k = 10$. The key reason behind this is that our generative approach gener-

ates a more diverse set of node mappings, which helps avoid sub-optimal solutions, whereas DiffGED-single's mappings tend to be highly correlated, leading to sub-optimal results. Moreover, even with iterative top-$k$ approach, it is interesting to note that DiffGED-single with $k = 100$ still achieves higher accuracy across all datasets compared to the results of GEDGNN and GEDIOT in Table 1, which highlights the effectiveness of our diffusion-based graph matching model DiffMatch.

**Efficiency.** Furthermore, as shown in Figure 6(d)-(f), the running time of DiffGED-single increases significantly faster than that of DiffGED as $k$ grows. This disparity arises from DiffGED-single's iterative top-$k$ node mapping strategy, whereas DiffGED benefits from parallelized node matching matrix generation and parallel node mapping extraction. Since both processes in DiffGED are parallelized, the impact of increasing $k$ on its running time remains minimal, underscoring its superior efficiency for larger $k$ values.

**Diversity.** Lastly, since multiple optimal edit paths often exist under a multimodal distribution, we evaluate edit paths diversity by computing the average number of distinct edit paths found per graph pair, where the number of edit operations is equal to the predicted minimum GED and the ground-truth GED, respectively, using $k = 100$. As demonstrated in Figure 7, our generative approach is capable of generating multiple distinct edit paths for both the predicted minimum GED and the ground-truth GED, while the iterative top-$k$ approach used in existing matching-based approaches is limited to generating only a few. This provides further evidence that our generative approach can generate diverse top-$k$ mappings, which enables us to effectively capture the multimodal distribution and avoid getting trapped in local optima. In contrast, the iterative approach used by existing frameworks produces highly correlated node mappings towards one mode, which limits its ability to capture the range of possible edit paths, thus could easily fall into sub-optimal results.

More experimental results (including more detailed ablation studies) can be found in Appendix D.

## 6. Conclusion

This paper presents DiffGED, a novel generative framework for GED computation and edit path generation. Our generative approach works by generating $k$ diverse node-matching matrices simultaneously through our diffusion-based graph matching model, DiffMatch, and then extracting the top-$k$ node mappings in parallel using a greedy algorithm. Extensive experiments on real-world datasets demonstrate that our generative method outperforms all existing approaches by generating diverse, high-quality edit paths with accuracy close to $100\%$, all within a short running time.

## Acknowledgments

Wei Huang is supported by the Commonwealth through an Australian Government Research Training Program Scholarship [DOI: https://doi.org/10.82133/C42F-K220]. Hanchen Wang is supported by DE250100226. Dong Wen is supported by ARC DP260100709 and ARC DE240100668.

## Impact Statement

This paper presents work whose goal is to advance the field of graph matching and databaase management. There are many potential societal consequences of our work, none of which we feel must be specifically highlighted here.

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

# A. Additional Related Work

**Graph matching.**   Graph matching is a problem closely related to GED and deep-learning based graph matching has garnered significant attention across various domains, particularly in image feature matching (Jiang et al., 2022b; Wang et al., 2023; Chen et al., 2019; Jiang et al., 2022a). However, a fundamental distinction between the two problems lies in the nature of their ground truth. In graph matching, the ground truth is typically unique and application-specific, whereas in GED, multiple valid ground truths may exist due to different possible edit paths leading to the same graph transformation. Additionally, while graph matching focuses on maximizing node correspondence with respect to a predefined ground truth, GED aims to determine the minimal sequence of edit operations required to transform one graph into another. Another key difference lies in the characteristics of the input graphs. In graph matching, the input graphs are often structurally similar, whereas in GED, they can differ significantly. As a result, existing graph matching methods struggle to perform well in GED computation.

**Diffusion model.**   Diffusion models have emerged as a powerful class of generative models, achieving remarkable success in image generation and setting new benchmarks for high-quality image synthesis (Ho et al., 2020; Dhariwal & Nichol, 2021; Sohl-Dickstein et al., 2015; Song & Ermon, 2019). These models progressively refine random noise into structured outputs through a learned denoising process, demonstrating superior performance over traditional generative approaches such as GANs and VAEs. The success of diffusion models in continuous domains has inspired extensions to discrete data, leading to the development of discrete diffusion models for structured tasks, such as text generation (Austin et al., 2021). Building on these advancements, discrete diffusion has been extensively applied to graph generation (Vignac et al., 2022; Haefeli et al., 2022; Sun & Yang, 2023), where it has shown great potential in downstream tasks such as molecule generation and combinatorial optimization. This success further motivates the exploration of diffusion-based methods for a broader range of graph-related problems beyond generation.

# B. Detailed Method

## B.1. Edit Path Extraction

The detailed algorithm for edit path extraction with linear time complexity $O(|V'| + |E| + |E'|)$ is illustrated in Algorithm 1.

## B.2. Training of DiffMatch

The training procedure of the denoising network in our DiffMatch is outlined in Algorithm 2. For a given graph pair $(G, G')$ sampled from the training data with its ground-truth matching matrix $M^0$, we first sample a time step $t$ from a uniform distribution. Next, we sample a noisy matching matrix $M^t$ from the $t$-step marginal. Finally, the denoising network is trained to minimize the binary cross-entropy loss between the predicted matching matrix $p_\theta(\widetilde{M}^0 | M^t, G, G')$ and the ground-truth node matching matrix $\widetilde{M}^0$.

## B.3. Inference of DiffMatch

Algorithm 3 illustrates the reverse process of DiffMatch during inference. During inference, starting from a noisy discrete node matching matrix $M^T$ randomly sampled from the Bernoulli distribution, each $M^{\tau_{i-1}}$ can be obtained from $p_\theta(M^{\tau_{i-1}} | M^{\tau_i}, G, G')$ via Bernoulli sampling. And for the last reverse step (i.e., $\tau_i = \tau_1$), we directly use $\hat{M} = p_\theta(M^0 | M^{\tau_1}, G, G')$ as the input of the node mapping extraction in phase 2.

---

**Algorithm 1** Edit Path Generation

---

**Input:** $G = (V, E, L)$, $G' = (V', E', L')$, node mapping $f$;

1: $EditCost \leftarrow 0$;
2: **for** each $v \in V$ **do**
3:     **if** $L(v) \neq L'(f(v))$ **then**
4:         $L(v) \leftarrow L'(v')$;
5:         $EditCost \leftarrow EditCost + 1$;
6:     **end if**
7: **end for**
8: **for** each $v' \in V' \setminus \{f(v) \mid v \in V\}$ **do**
9:     Create a new $v$;
10:     $f(v) \leftarrow v'$ and $L(v) \leftarrow L'(v')$;
11:     $V \leftarrow V \cup \{v\}$;
12:     $EditCost \leftarrow EditCost + 1$;
13: **end for**
14: **for** each $(v, u) \in E$ **do**
15:     **if** $(f(v), f(u)) \in E'$ **then**
16:         $E \leftarrow E \setminus \{(v, u)\}$;
17:         $EditCost \leftarrow EditCost + 1$;
18:     **end if**
19: **end for**
20: **for** each $(v', u') \in E'$ **do**
21:     **if** $(f^{-1}(v), f^{-1}(u)) \notin E$ **then**
22:         $E \leftarrow E \cup \{(f^{-1}(v), f^{-1}(u))\}$;
23:         $EditCost \leftarrow EditCost + 1$;
24:     **end if**
25: **end for**
26: **return** $EditCost$;

---

**Algorithm 2** DiffMatch Training Procedure

---

**Input:** Graph pair $(G, G')$, Ground-truth node matching matrix $M^0$;

1: Sample $t \sim Uniform(1, ..., T)$;
2: Sample $M^t \sim q(M^t | M^0)$;
3: Take gradient step on $BCELoss(p_\theta(\widetilde{M}^0 | M^t, G, G'), M^0)$ via Equation 1;

---

## B.4. Anisotropic Graph Neural Network

For each layer $l$ of our denoising network, the Anisotropic Graph Neural Network (AGNN) can be represented as follows:

$$
\begin{aligned}
\hat{\boldsymbol{h}}_{vv'}^l &= \boldsymbol{W}_1^l \boldsymbol{h}_{vv'}^{l-1}, \quad \hat{\boldsymbol{h}}_{v'v}^l = \boldsymbol{W}_1^l \boldsymbol{h}_{v'v}^{l-1} \\
\tilde{\boldsymbol{h}}_{vv'}^l &= \boldsymbol{W}_2^l \hat{\boldsymbol{h}}_{vv'}^l + \boldsymbol{W}_3^l \hat{\boldsymbol{h}}_v^l + \boldsymbol{W}_4^l \hat{\boldsymbol{h}}_{v'}^l \\
\tilde{\boldsymbol{h}}_{v'v}^l &= \boldsymbol{W}_2^l \hat{\boldsymbol{h}}_{v'v}^l + \boldsymbol{W}_3^l \hat{\boldsymbol{h}}_{v'}^l + \boldsymbol{W}_4^l \hat{\boldsymbol{h}}_v^l \\
\boldsymbol{h}_{vv'}^l &= \hat{\boldsymbol{h}}_{vv'}^l + \text{MLP}^l(\text{ReLU}(\text{GN}_{MM^\top}(\tilde{\boldsymbol{h}}_{vv'}^l)) + \boldsymbol{W}_5^l \boldsymbol{h}_t) \\
\boldsymbol{h}_{v'v}^l &= \hat{\boldsymbol{h}}_{v'v}^l + \text{MLP}^l(\text{ReLU}(\text{GN}_{MM^\top}(\tilde{\boldsymbol{h}}_{v'v}^l)) + \boldsymbol{W}_5^l \boldsymbol{h}_t) \\
\boldsymbol{h}_v^l &= \hat{\boldsymbol{h}}_v^l + \text{ReLU}(\text{GN}_{GG'}(\boldsymbol{W}_6^l \hat{\boldsymbol{h}}_v^l + \sum_{v' \in V'} \boldsymbol{W}_7^l \hat{\boldsymbol{h}}_{v'}^l \odot \sigma(\tilde{\boldsymbol{h}}_{vv'}^l))) \\
\boldsymbol{h}_{v'}^l &= \hat{\boldsymbol{h}}_{v'}^l + \text{ReLU}(\text{GN}_{GG'}(\boldsymbol{W}_6^l \hat{\boldsymbol{h}}_{v'}^l + \sum_{v \in V} \boldsymbol{W}_7^l \hat{\boldsymbol{h}}_v^l \odot \sigma(\tilde{\boldsymbol{h}}_{v'v}^l)))
\end{aligned}
\tag{5}
$$

where $\boldsymbol{W}_1^l, \boldsymbol{W}_2^l, \boldsymbol{W}_3^l, \boldsymbol{W}_4^l, \boldsymbol{W}_5^l, \boldsymbol{W}_6^l, \boldsymbol{W}_7^l$ are learnable parameters at layer $l$, MLP$^l$ denotes multi-layer perceptron at layer $l$, GN$_{MM^\top}$ is the graph normalization (Cai et al., 2021) over all node matching pairs in both $M^t$ and $M^{t^\top}$, GN$_{GG'}$ is the graph normalization over all nodes in both $G$ and $G'$, and $\sigma$ is the sigmoid activation.

---

**Algorithm 3** Sampling from DiffMatch

---

**Input:** Graph pair $(G, G')$, Random node matching matrix $M^T$;
1: **for** $\tau_i = \tau_S$ to $\tau_1$ **do**
2:   **if** $\tau_i \neq \tau_1$ **then**
3:     $M^{\tau_{i-1}} \sim p_\theta(M^{\tau_{i-1}} | M^{\tau_i}, G, G')$;
4:   **else**
5:     $\hat{M} \leftarrow p_\theta(M^0 | M^{\tau_1}, G, G')$;
6:   **end if**
7: **end for**
8: **return** $\hat{M}$;

---

**Algorithm 4** Greedy Node Mapping Extraction

---

**Input:** $i$-th node matching matrix $\hat{M}_i \in \mathbb{R}^{|V| \times |V'|}$;
**Output:** $i$-th node mapping $f_i$;
1: Initialize $f_i \leftarrow \emptyset$ ;
2: **for** $n \leftarrow 1$ to $|V|$ **do**
3:   select $(v, v')$ with the maximum value in $\hat{M}_i$;
4:   $f_i \leftarrow f_i \cup \{(v, v')\}$;
5:   set all elements in $v$-th row of $\hat{M}_i$ to $-\infty$;
6:   set all elements in $v'$-th column of $\hat{M}_i$ to $-\infty$;
7: **end for**
8: **return** $f_i$;

---

### B.5. Phase 2: Node Mapping Extraction

Given a predicted node matching matrix $\hat{M}_i$, Algorithm 4 outlines the overall greedy procedure to extract top-1 node mapping from $\hat{M}_i$.

## C. Detailed Experimental Settings

### C.1. Datasets

**Dataset construction.** We conduct experiments over three popular real-world GED datasets: AIDS700 (Bai et al., 2019), Linux (Wang et al., 2012; Bai et al., 2019) and IMDB (Bai et al., 2019; Yanardag & Vishwanathan, 2015). Specifically, AIDS contains 700 labeled graphs with up to 10 nodes, Linux contains 1000 unlabeled graphs with up to 10 nodes, and IMDB contains 1500 unlabeled graphs with up to 89 nodes. We obtain the ground-truth edit path (node mappings) from (Piao et al., 2023). However, the ground-truth GED and edit paths are often computationally expensive to obtain for graph pairs with at least one graph has more than 10 nodes. To handle this, we follow the same strategy as described in (Piao et al., 2023) to generate synthetic graphs for IMDB dataset. Specifically, for each graph $G$ with more than 10 nodes, synthetic graphs are generated by randomly applying $\Delta$ edit operations to $G$, these random edit operations are used as an approximation of the ground-truth edit path and $\Delta$ is used as an approximate of ground-truth GED. For graphs with more than 20 nodes, $\Delta$ is randomly distributed in $[1, 10]$, for graphs with more than 10 nodes and less than 20 nodes, $\Delta$ is randomly distributed in $[1, 5]$.

**Dataset split.** For each dataset, we split $60\%$, $20\%$, and $20\%$ of all the graphs as training set, validation set, and testing set, respectively. To form training pairs, each training graph with no more than 10 nodes is paired with all other training graphs with no more than 10 nodes, each training graph with more than 10 nodes is paired with 100 synthetic graphs. To construct validation and testing pairs, we follow the standard setting adopted in previous works. Specifically, each validation/testing graph with no more than 10 nodes is paired with 100 randomly selected training graphs containing no more than 10 nodes, while each graph with more than 10 nodes is paired with 100 synthetic graphs. In addition to the standard cross-train-test pair construction, we further construct intra-test pairs by pairing each testing graph with 100 randomly selected testing/synthetic graphs.

*Table 3.* Overall Performance on IMDB cross-train-test pairs. IMDB-small refers to training set that only contains real small graph pairs. IMDB-mix refers to training set that contains a combination of real small graph pairs and synthetic large graph pairs.

| Training set | Models | MAE ↓ | Accuracy ↑ | $\rho$ ↑ | $\tau$ ↑ | p@10 ↑ | p@20 ↑ | Time(s) ↓ |
|---|---|---|---|---|---|---|---|---|
| | GEDGNN | 7.943 | 77.1% | 0.844 | 0.815 | 88.2% | 87.6% | 0.48253 |
| IMDB-small | GEDIOT | 7.761 | 76.8% | 0.86 | 0.827 | 90.5% | 89.9% | 0.473 |
| | DiffGED | 5.789 | 83% | 0.892 | 0.874 | 90.1% | 90.8% | 0.14923 |
| | GEDGNN | 2.469 | 85.5% | 0.898 | 0.879 | 92.4% | 92.1% | 0.42428 |
| IMDB-mix | GEDIOT | 2.822 | 84.5% | 0.9 | 0.878 | 92.3% | 92.7% | 0.41959 |
| | DiffGED | 0.937 | 94.6% | 0.982 | 0.973 | 97.5% | 98.3% | 0.15105 |

## C.2. Details of Baseline Methods

**Traditional methods.** For traditional approximation methods, we compare our DiffGED with: (1) **Hungarian** (Riesen & Bunke, 2009) and (2) **VJ** (Bunke et al., 2011). Note that we do not include traditional optimization-based methods such as F2 (Lerouge et al., 2017), as we do not have access to the Gurobi solver required for these methods.

**Hybrid methods.** For A* search-based hybrid methods, we compare with: (1) **Noah** (Yang & Zou, 2021) proposed using a pre-trained Graph Path Network (GPN) as the heuristic for A* beam search; (2) **GENN-A*** (Wang et al., 2021) introduced a Graph Edit Neural Network (GENN) to guide A* search by dynamically predicting the edit costs of unmatched subgraphs; (3) **MATA*** (Liu et al., 2023) proposed to prune the search space of A* search by extracting top-$k$ candidate matches for each node from two predicted node matching matrices.

For matching-based hybrid methods, we compare with: (1) **GEDGNN** (Piao et al., 2023) predicts a single deterministic node matching matrix, then iteratively extracts top-$k$ node mappings and edit paths; (2) **GEDIOT**(Cheng et al., 2025) follows the same approach as GEDGNN and further improves the prediction of node matching matrix via optimal transport.

Note that, since our evaluation is based on feasible discrete GED values derived from actual edit paths, we do not include regression-based methods for comparison.

## C.3. Implementation Details

During training of our DiffMatch, we set the number of time steps $T$ to $1,000$ with linear noise schedule, where $\beta_0 = 10^{-4}$ and $\beta_T = 0.02$. For the reverse denoising process during testing, we set the number of time steps $S$ to 10 with linear denoising schedule, and we generate $k = 100$ node matching matrices in parallel for each testing graph pair.

For our denoising network, we set the number of layers to 6, the output dimension of each layer is 128, 64, 32, 32, 32, 32, respectively. We train it for 200 epochs with batch size of 128, we adopt Adam optimizer (Kingma, 2014) with learning rate of 0.001 and weight decay of $5 \times 10^{-4}$.

All experiments are conducted using Nvidia Geforce RTX3090 24GB and Intel i9-12900K with 128GB RAM.

# D. More Experimental Results

## D.1. Generalization to Large Graphs

In real-world scenarios, obtaining ground-truth node mappings for large graph pairs is often impractical. To evaluate the generalization ability of DiffGED under such conditions, we modify the training setup. Instead of training each method on a combination of real small graph pairs and synthetic large graph pairs from IMDB, we train each method exclusively on real small graph pairs from IMDB. However, the testing set still consists of a combination of real small graph pairs and synthetic large graph pairs. Table 3 presents the overall performance of DiffGED, GEDGNN and GEDIOT when trained on real small graph pairs. As observed, the accuracy of both DiffGED, GEDGNN and GEDIOT degrades, primarily because the testing graph pairs differ from the training graph pairs not only in graph size but also in distribution, due to the presence of synthetic graph pairs in the testing set, as these synthetic graphs differ from real graph pairs. Despite this challenge, DiffGED still outperforms GEDGNN and GEDIOT, achieving higher accuracy.

*Table 4.* Overall performance on cross-train-test pairs under unsupervised training setting.

| Datasets | Models | MAE ↓ | Accuracy ↑ | $\rho$ ↑ | $\tau$ ↑ | p@10 ↑ | p@20 ↑ |
|---|---|---|---|---|---|---|---|
| AIDS700 | GEDGNN | 1.279 | 44.6% | 0.809 | 0.712 | 82.6% | 83.7% |
| | DiffGED | **0.088** | **92.6%** | **0.984** | **0.969** | **99.1%** | **99.1%** |
| Linux | GEDGNN | 0.078 | 96.3% | 0.982 | 0.971 | 98.2% | 98.7% |
| | DiffGED | **0.01** | **99.5%** | **0.997** | **0.995** | **100%** | **99.8%** |
| IMDB | GEDGNN | 2.536 | 85.2% | 0.896 | 0.876 | 92.0% | 91.9% |
| | DiffGED | **1.019** | **94%** | **0.999** | **0.97** | **96.1%** | **97%** |

## D.2. Performance Evaluation under Unsupervised Training Setting

In this section, we evaluate the performance of DiffGED in the unsupervised training setting, where no ground-truth labels are available during training. Specifically, we train DiffGED and other matching-based methods (i.e., GEDGNN) using the latest unsupervised training strategy, GEDRanker (Huang et al., 2025), which is particularly designed for matching-based hybrid approaches. Notably, GEDIOT is not included in this unsupervised evaluation, as it shares the same architecture as GEDGNN, and applying GEDRanker to GEDIOT would naturally become equivalent to adopting GEDGNN with GEDRanker under the same unsupervised training setting.

As shown in Table 4, DiffGED still consistently outperforms GEDGNN in the unsupervised training setting. And surprisingly, DiffGED trained without any supervision also surpasses all baseline methods trained in the supervised training setting reported in Table 1, demonstrating the strong effectiveness of our proposed DiffGED.

## D.3. Detailed Ablation Studies

*Table 5.* Ablation study on cross-train-test pairs.

| Datasets | Models | MAE ↓ | Accuracy ↑ | $\rho$ ↑ | $\tau$ ↑ | p@10 ↑ | p@20 ↑ | Time(s) ↓ |
|---|---|---|---|---|---|---|---|---|
| AIDS700 | GEDGNN(Noise) | 1.265 | 42.3% | 0.815 | 0.718 | 80.8% | 82.6% | 0.0079 |
| | Random Sampling | 4.858 | 4.7% | 0.564 | 0.459 | 52.2% | 63.2% | 0.00742 |
| | DiffGED(w/o diffusion) | 1.618 | 46.7% | 0.732 | 0.629 | 82.4% | 81.1% | 0.01179 |
| | DiffGED(VAE) | 1.435 | 48.7% | 0.754 | 0.653 | 84.3% | 81.9% | 0.01135 |
| | GEDGNN(AGNN) | 0.736 | 66.7% | 0.884 | 0.812 | 94% | 93.1% | 0.39112 |
| | GEDGNN | 1.098 | 52.5% | 0.845 | 0.752 | 89.1% | 88.3% | 0.39448 |
| | DiffGED | 0.022 | 98% | 0.996 | 0.992 | 99.8% | 99.7% | 0.0763 |
| Linux | GEDGNN(Noise) | 0.102 | 95.4% | 0.985 | 0.973 | 98.4% | 98.9% | 0.00769 |
| | Random Sampling | 2.869 | 14.2% | 0.746 | 0.659 | 73.9% | 79.4% | 0.00735 |
| | DiffGED(w/o diffusion) | 0.743 | 74.7% | 0.887 | 0.839 | 96.4% | 95.8% | 0.01117 |
| | DiffGED(VAE) | 0.828 | 71.9% | 0.875 | 0.821 | 96.1% | 95.3% | 0.01088 |
| | GEDGNN(AGNN) | 0.061 | 97.4% | 0.992 | 0.987 | 99.6% | 99.5% | 0.13164 |
| | GEDGNN | 0.094 | 96.6% | 0.979 | 0.969 | 98.9% | 99.3% | 0.12863 |
| | DiffGED | 0.0 | 100% | 1.0 | 1.0 | 100% | 100% | 0.06982 |
| IMDB | GEDGNN(Noise) | 13.885 | 75.6% | 0.805 | 0.788 | 84.7% | 84.6% | 0.009 |
| | Random Sampling | 26.143 | 59.9% | 0.641 | 0.627 | 69.2% | 73.4% | 0.00803 |
| | DiffGED(w/o diffusion) | 0.832 | 93.3% | 0.942 | 0.93 | 98.6% | 96.8% | 0.01944 |
| | DiffGED(VAE) | 0.74 | 94.3% | 0.963 | 0.949 | 98.9% | 97.9% | 0.01668 |
| | GEDGNN(AGNN) | 1.766 | 89.1% | 0.903 | 0.89 | 93.9% | 92.8% | 0.41387 |
| | GEDGNN | 2.469 | 85.5% | 0.898 | 0.879 | 92.4% | 92.1% | 0.42428 |
| | DiffGED | 0.937 | 94.6% | 0.982 | 0.973 | 97.5% | 98.3% | 0.15105 |

**Do we really need generative modeling?** To evaluate the necessity of our proposed generative modeling of graph edit distance computation, we construct a variant model, GEDGNN(Noise), in which the generative modeling is removed. Instead of learning to generate node matching matrices via generative model, this variant injects random noise into the deterministic matching matrix predicted by GEDGNN to produce multiple perturbed matching matrices. We use this variant to examine whether such a simple approach can enhance GEDGNN and serve as an effective alternative to DiffGED.

As shown in Table 5, this approach fails to improve the performance of GEDGNN and remains significantly inferior to our generative approach, DiffGED. This is because: (1) when the injected noise is small, it does not substantially change

the predicted base matching matrix (i.e., the probability distribution), resulting in perturbed matching matrices that are highly correlated with the original one; consequently, the node mappings extracted from these matrices also remain highly correlated; (2) when the injected noise is large, it may significantly alter (flatten) the matching matrix, but in an uncontrolled manner (i.e., not conditioned on the input graph pair), causing the resulting matrices to become essentially random and semantically meaningless. To further validate this, we construct another variant, Random Sampling, which directly extracts node mappings from multiple random node matching matrices. As shown in Table 5, this unconditional random approach performs poorly.

In contrast, our DiffGED takes different random noise as inputs and learns to denoise them in a controlled manner (i.e., conditioned on the underlying graph pair) via the generative matching model DiffMatch, guiding the random matrices toward diverse and high-quality node matching matrices. Therefore, neither GEDGNN(Noise) nor Random Sampling can serve as effective alternatives to our generative approach, DiffGED.

**Do we really need diffusion model?** The core idea of the proposed framework is to generate diverse, high-quality node matching matrices through an iterative reverse process of the diffusion model. To assess the effectiveness of the generative diffusion model in DiffMatch, we introduce a one-shot generative variant model, DiffGED(w/o diffusion), which takes a graph pair and a randomly initialized node matching matrix as input and directly predicts the clean node matching matrix, followed by greedy node mapping extraction. In this setup, we remove the time step component from the denoising network. During training, DiffGED(w/o diffusion) is also provided with a random node matching matrix instead of a noisy node matching matrix sampled from the forward diffusion process. Notably, this approach can also be viewed as removing the discriminator from GANs and directly training the generator of GANs to recover the ground-truth. For completeness, we further create another alternative generative model, DiffGED (VAE), which replaces the diffusion model with a VAE.

Table 5 presents the overall performance of DiffGED(w/o diffusion) and DiffGED(VAE). Notably, both alternative generative models perform poorly, with performance even worse than GEDGNN and GEDIOT on the AIDS and Linux datasets.

From a solution quality perspective, both DiffGED(w/o diffusion) and DiffGED(VAE) attempts to generate a high-quality node matching matrix in a single step from random noise, making the learning task extremely challenging. In contrast, the reverse diffusion process of the diffusion model gradually denoises the random node matching matrix step by step, ensuring that each step only requires minor corrections. This progressive refinement breaks down this complex denoising task into simpler iterative steps, thereby producing higher-quality node matching matrices.

**Anisotropic Graph Neural Network.** Instead of computing only node embeddings and then using their inner product to predict node matching probabilities, our matching-aware denoising network leverages the Anisotropic Graph Neural Network (AGNN) to directly compute node pair embeddings, enabling a more expressive prediction of node matching probabilities.

To evaluate the effectiveness of AGNN, we create a variant of GEDGNN, GEDGNN(AGNN), that replaces its Cross Matrix Module with AGNN (without time steps). Moreover, we initialize a fixed node matching matrix filled with ones as input of GEDGNN(AGNN). We choose to create a variant of GEDGNN rather than creating a variant of DiffMatch by replacing AGNN with the Cross Matrix Module. This is because DiffMatch requires a noisy node matching matrix as input, but the Cross Matrix Module of GEDGNN ($\text{MLP}([h_v^\top W_1 h_{v'}, ..., h_v^\top W_c h_{v'}])$) cannot incorporate such noisy information when computing node matching probabilities. This limitation makes Cross Matrix Module unsuitable for direct integration into DiffMatch, leading us to use GEDGNN(AGNN) as the evaluation model for AGNN instead.

The overall performance of GEDGNN(AGNN) is presented in Table 5. The performance of GEDGNN increased significantly by incorporating AGNN, demonstrating that AGNN effectively enhances the model's ability to predict node matching probabilities by directly computing expressive node pair embeddings.

**Varying Reverse Denoising Steps During Inference.** During inference, DiffMatch denoises noisy node matching matrices through $S$ reverse steps. To assess the impact of the number of reverse denoising steps on DiffGED's performance, we evaluate DiffGED using different values of $S$, specifically $S = [20, 10, 5, 4, 3, 2, 1]$. Figure 8 presents the performance comparison across different values of $S$. The results indicate that when $S > 2$, the accuracy and MAE of DiffGED do not vary a lot. However, when $S \leq 2$, accuracy drops significantly while MAE increases. In particular, at $S = 1$, DiffGED becomes a one-shot model, suffering from the same limitations as DiffGED(w/o diffusion), leading to similarly poor performance. Moreover, when $S$ is doubled, the running time of DiffGED almost doubles as well, as the majority of its computational cost comes from denoising the node matching matrix at each reverse step.

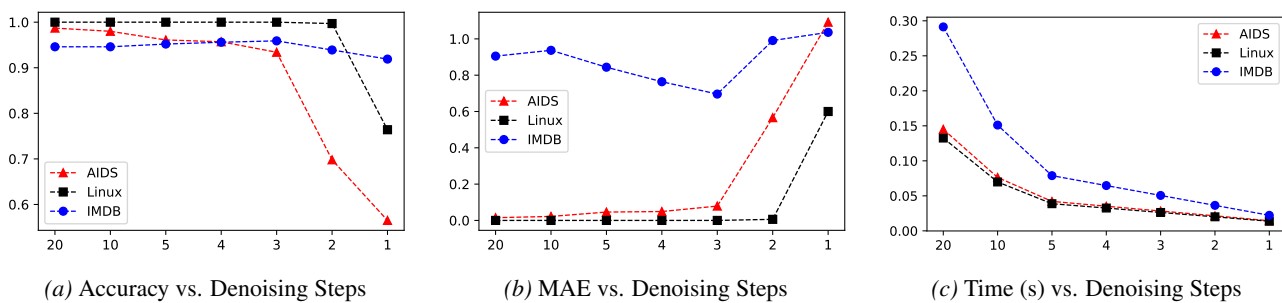

*(a)* Accuracy vs. Denoising Steps     *(b)* MAE vs. Denoising Steps     *(c)* Time (s) vs. Denoising Steps

*Figure 8.* Performance comparison across different reverse denoising steps during inference.

*Table 6.* Evaluation on Node Mapping Extraction Strategy.

| Datasets | Models | MAE ↓ | Accuracy ↑ | Extraction Time(s) ↓ |
|---|---|---|---|---|
| AIDS700 | DiffGED | 0.022 | 98% | 0.00043 |
| | DiffGED(Hungarian) | 0.021 | 98.1% | 0.0035 |
| Linux | DiffGED | 0.0 | 100% | 0.00036 |
| | DiffGED(Hungarian) | 0.0 | 100% | 0.00345 |
| IMDB | DiffGED | 0.937 | 94.6% | 0.00068 |
| | DiffGED(Hungarian) | 0.918 | 94.7% | 0.00367 |

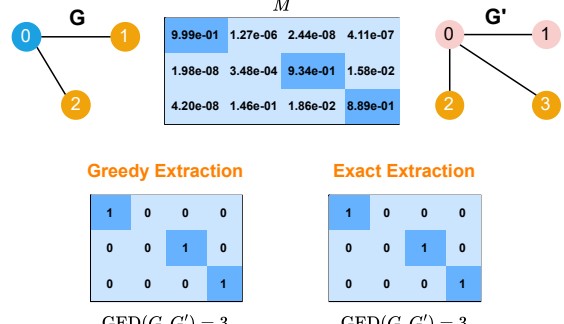

*Figure 9.* Greedy vs. Exact Node Mapping Extraction.

**Greedy vs. Exact Node Mapping Extraction.** To evaluate the effectiveness and efficiency of greedy node mapping extraction, we introduce a variant model, DiffGED(Hungarian), which replaces the greedy extraction method with the exact Hungarian algorithm (Kuhn, 1955). As shown in Table 6, DiffGED with greedy node mapping extraction achieves nearly identical accuracy and MAE to DiffGED(Hungarian) across all datasets, while significantly reducing the computational cost of node mapping extraction. This improvement stems from the fact that DiffMatch generates a high-quality sparse node matching matrix, where most elements in each row and column are close to 0, with only a few elements close to 1. This sparsity enables the greedy extraction method to retrieve node mappings comparable to those obtained by the exact Hungarian algorithm while being much faster. To better illustrate this, we show a simple example graph pair in Figure 9, where $\hat{M}$ represents the node matching matrix predicted by DiffMatch. We can see that the predicted $\hat{M}$ is both high-quality and sparse, leading to identical extracted node mappings under both the greedy and Hungarian strategies, resulting in $GED(G, G') = 3$.

## E. Additional Datasets & Baselines

**Additional datasets.** In this section, we further evaluate DiffGED on two additional datasets, ZINC and Code2 (Gómez-Bombarelli et al., 2018; Hu et al., 2020). We use the datasets and ground-truth GED labels processed by (Pellizzoni et al., 2025). Notably, the processed datasets are already de-leaked, and the graphs in the ZINC dataset are edge-labeled, enabling a more comprehensive evaluation of DiffGED under a more general and advanced GED computation setting.

Table 7 presents the performance of DiffGED on the ZINC and Code2 intra-test graph pairs. The results show that DiffGED still consistently achieves promising performance and outperforms the baseline matching-based hybrid approaches, GEDGNN and GEDIOT. Notably, on the edge-labeled ZINC dataset, GEDGNN and GEDIOT exhibit poor performance, whereas DiffGED still performs well. This further demonstrates the effectiveness of DiffGED under a more general and challenging GED computation setting.

**Comparison with LLM-designed traditional heuristics.** Recently, with the rapid advancement of large language

*Table 7.* Overall Performance on ZINC and Code2 intra-test graph pairs.

| Datasets | Models | MAE ↓ | Accuracy ↑ | Time(s) ↓ |
|---|---|---|---|---|
| ZINC | GEDGNN | 5.198 | 6.6% | 0.70134 |
| | GEDIOT | 5.882 | 4.1% | 0.71695 |
| | DiffGED (ours) | **0.387** | **70.8%** | **0.12902** |
| Code2 | GEDGNN | 1.237 | 62.6% | 1.05171 |
| | GEDIOT | 2.588 | 36.1% | 1.05512 |
| | DiffGED (ours) | **0.04** | **97%** | **0.21924** |

*Table 8.* Comparison with LLM-designed traditional heuristics on intra-test graph pairs.

| Datasets | Models | MAE ↓ | Accuracy ↑ | Time(s) ↓ |
|---|---|---|---|---|
| AIDS | Grail | 0.204 | 83.7% | **0.005** |
| | DiffGED (ours) | **0.024** | **96.4%** | 0.07546 |
| Linux | Grail | 0.01 | 99.5% | **0.00333** |
| | DiffGED (ours) | **0.0** | **100%** | 0.06901 |
| IMDB | Grail | **0.025** | **99.3%** | 0.21144 |
| | DiffGED (ours) | 0.932 | 94.6% | **0.15107** |
| ZINC | Grail | 1.777 | 20.4% | **0.012** |
| | DiffGED (ours) | **0.387** | **70.8%** | 0.12902 |
| Code2 | Grail | 0.635 | 63.9% | **0.045** |
| | DiffGED (ours) | **0.04** | **97%** | 0.21924 |

models (LLMs), there has been growing interest in leveraging LLMs for the design of traditional heuristics. In particular, Grail (Verma et al., 2025) proposes to employ LLMs to generate heuristics for GED computation. In this section, we further compare DiffGED with the heuristics discovered by Grail.

As shown in Table 8, DiffGED achieves better MAE and Accuracy on ZINC, Code2, AIDS, and Linux datasets. Notably, the ground-truth GEDs for IMDB evaluation are approximated since it contains large graphs. Therefore, the results on IMDB only reflect the quality of approximation relative to the approximated ground-truth, rather than the exact GED values (i.e., the approximated ground-truth ≥ the true GED). Moreover, while the heuristics generated by Grail are efficient on small graphs, their running time increases significantly with graph size and becomes slower than DiffGED on large graphs (e.g., IMDB). In contrast, DiffGED exhibits more stable computational efficiency across different graph scales. Finally, Grail relies heavily on LLM calls to generate heuristics, which is computationally expensive. In contrast, DiffGED can be trained using a single RTX 3090 GPU, making it significantly more cost-effective.

## F. Potential Future Directions

This work introduces a novel generative formulation for graph matching using diffusion models. Although our diffusion-based graph matching framework demonstrates promising performance compared with previous methods, it is worth noting that diffusion models require multiple denoising steps during inference, making them less efficient than previous approaches when generating a single node matching matrix. One potential direction for future improvement is to accelerate the diffusion process through consistency training (Song et al., 2023; Song & Dhariwal, 2024), which can reduce the number of denoising steps during inference, or through other generative paradigms (Lipman et al., 2022; Liu et al., 2022; Geng et al., 2026).

Moreover, existing approaches for GED approximation, including our DiffGED, require estimating matching matrix of size $|V||V'|$. This quadratic complexity may limit the scalability of current methods on large-scale graphs, an issue that remains largely unexplored in prior work. Therefore, another promising future direction is to effectively reduce the size of the matching matrix by designing heuristic-based pruning strategies. And for our diffusion-based approach, techniques such as latent diffusion (Rombach et al., 2022; Yang et al., 2024) could also be employed to reduce the dimension of matching matrix for intermediate denoising steps.

Furthermore, one of the major applications of GED computation is graph similarity search in graph databases. In real-world settings, graph databases are often dynamically evolving, with new graphs continuously being added over time. Such scenarios may require models to be frequently updated in order to adapt to newly observed graph patterns and distributions (e.g., from different domains). However, naively retraining the model on newly added graphs may lead to catastrophic forgetting, causing performance degradation on previously seen graphs. Therefore, another potential future research direction is to incorporate transfer learning or incremental learning techniques (Weiss et al., 2016; Van de Ven et al., 2022; Zhang et al., 2024; Zhao et al., 2024; 2025) into GED approximation frameworks, enabling the model to incrementally learn from newly arriving graphs while maintaining strong performance on previously learned graph distributions in the real-world database setting.

