# OpenReview forum: "Towards Generative Graph Matching for Graph Edit Distance Computation"
_ICML.cc/2026/Conference — ICML 2026 regular_

### Official Review · Reviewer_z28D · 2026-02-20

**Soundness:** 2
**Presentation:** 2
**Significance:** 3
**Originality:** 3
**Overall Recommendation:** 4
**Confidence:** 4

**Summary:**

The author's propose DiffGED, a discrete diffusion framework which overcomes current limitations with neural-based Graph Edit Distance frameworks in order to achieve state-of-the-art performance. Numerous details about design decisions for more effectively utilizing node-mappings with generative diffusion are given (Anisotropic GNNs, graph pairs, and separate generation and mapping phases), where authors delineate DiffGED from existing A*-star based and hybrid systems. Experiments on well-established benchmarks (AIDS700, Linux, IMDB) demonstrate that DiffGED shows significant accuracy (and other metric) improvements of 36.5%, 3.3%, and 10.1% over previous SOTA baselines. The authors also details time trade-offs between a specialized DiffGED-single variant to further indicate practical applicability of the system.

**Compliance With Llm Reviewing Policy:**

Affirmed.

**Final Justification:**

Mostly addressed my main concerns.

**Key Questions For Authors:**

* Would it be feasible to apply MMD, EMD, or TVD across structural characteristics to measure the diversity of node-mappings generated by DiffGed versus other baselines? Or would it require a separate metric? This is crucial to provide empirical evidence on the claim that DiffGED generates diverse node-mappings and could provide further insights into DiffGED's practical effectiveness.
* Why is there no discussion on the quadratic time-complexity of discrete graph diffusion? Given that DiffGED is centered around GED-approximation it is a reasonable oversight but further application and scalability to larger datasets mean this is an important limitation to include within paper discussion.
* Was the inclusion of plain-speech explanations mixed with equations a space-saving measure for the review submission? If so, it may seem prudent to prune repeated claims or tighten writing in order to include references to algorithms to detail technical contributions or include discussion on real-world applications of DiffGED
Limitations: No, a central limitation of discrete/categorical graph diffusion is it's quadratic time-complexity across nodes and edges. Further discussion on how this affects DiffGED or potential tricks for allowing DiffGED to bypass this problem would enable future researchers to understand a central limitation to this paper's contribution.

**Limitations:**

yes

**Strengths And Weaknesses:**

Strengths:
-------------

* The practical downstream effectiveness of DiffGED is sound, Table 1 and Figure 7 in the main body of the paper indicate DiffGED's strong performance for limited time trade-off compared to MATA. This is further supported by Tables 2 through 6 in the Appendix. However, the re-iterated claim that DiffGED breaks free from the pre-existing limitation of extracting node mappings 'from a single deterministic node matching matrix' by generating more diverse node-mappings has no direct evidence beyond it's downstream performance, further evaluation on the diversity of these node-mappings would make the soundness of DiffGED absolute and provide a valuable insight to the community of the application of discrete graph generation for GED computation. The lack of direct texting on the diversity of DiffGED-generated node-mapping is the largest contributing factor to this review's current score.
* Much of the writing is the submission is clear, where the authors take great care to provide technical details to better enable reproducibility without requiring a code implementation (which they also include). However, the writing suffers from numerous grammatical issues which make the narrative harder to follow, details included below on a Line-by-Line basis.

Weaknesses:
-----------------

* This may be due to space limitations but much of technical discussion 'Phase 1: DiffMatch' and 'Phase 2: Node Mapping Extraction' is centered around plain speech explanation and equations, when it would do well for readers to include direct references to Algorithms 1-4.
* Graph Edit Distance (GED) calculation is a fundamental computer science problem, the NP-hard time complexity of exact GED solutions means better GED-approximations speak for itself as technical contributions. Despite this, the paper is centered exclusively around the technical components of DiffGED; it would do well for future researchers if there was some form of discussion on the practical, real-world applications of GED-approximation and what DiffGED could enable for this given its strong performance.
* DiffGED provides a technically-nuanced perspective on approaching GED-approximation by applying graph generation within a two-phase approximation framework. The originality in the submission is not within the discrete graph generation process itself but the application of DiffMatch for node matching to provide high-quality samples for the greedy edit path extraction. The authors make solid case for the what and the why within their implementation, further supported by ablation studies in the Appendix. However, there is a trade-off where discrete graph diffusion imposes a quadratic time complexity for the node-matching process (which appears evident in the time trade-off between MATA), discussion about this in the appendix is important for future applications of DiffGED to understand critical limitations of the categorical diffusion process itself and how it affects DiffGED.

Other:
-------
A non-exhaustive list of suggested line-edits to improve readability and presentation of the paper.
* Line 80-82 (left column): "It is clear to see that the top-6 node mappings are extracted from the predicted matching matrix are high correlated, and unfortunately, they are sub-optimal with the derived GED significantly larger than the ground-truth GED". -- Almost a Run-On Sentence, could be spliced or shortened to improve readability.
* Line 94-96 (right): "To improve scalability, traditional matching-based approaches proposed to construct a node edition cost matrix, then model GED as a bipartite node matching problem and solve by either Hungarian or VJ algorithm in polynomial time." -- Use of passive voice, who is improving scalability?
* Line 149 (left): "Graph Edit Distance (GED). (Sanfeliu & Fu, 1983)" -- include citation with problem definition instead of subsection title?
* Line 136-138 (right): "To obtain top-k node mappings, our DiffGED proposes a two-phase approach as shown in Figure 3." -- add comma, does DiffGED propose the two-phase approachs or do you the authors? 'DiffGED acts in two-phases' reads better for this sentence.
* Line 160-162 (right): "Once trained, they can only produce a fixed number of node matching matrices (often just one), and this requires a corresponding number of prediction heads in the network architecture, which consequently increases the number of unnecessary network parameters." -- begging the question, why are the network parameters unnecessary, because they number of prediction heads is deterministic. A bit hard to track and define so absolutely given the scope of the problem presented and the presentation, needs direct link as to why this is a problem..Citations? Appendices?
* Line 190-192 (left): "generative approach" --> "a generative approach".
* Line 191-192 (left): re-proposing DiffMatch..maybe re-word the sentence to describe more of the novelty of the mechanism.
* Line 190-192 (right): "Furthermore, each refinement steps introduces stochasticity, which further reduces the correlation between generate node matching matrices and enhances the model's ability to produce diverse node matching matrices" --- reclaiming what was said before
* Line 239 (right): "we rewritten the posterior as follows:" --> "we rewrite the posterior as follows:"
* Lines 305-306 (left): "The key advantage of AGNN is it ability to directly updating" --> "direct updating"
* Lines 415-417 (right): "This further evident" --> "This evidence"

---

> ### Author Rebuttal · Authors · 2026-03-30
>
> ### **Presentation**
>
> Thanks for the suggestion. We will include direct references and the suggested line-edits to improve readability.
>
> ---
>
> ### **Diversity measurement**
>
> In our paper, diversity refers to the number of distinct optimal or high-quality node mappings that a method can discover.
> To quantify this, in Figure 7 (please see the last paragraph of Section 5.2) we evaluated the average number of distinct node mappings identified by our generative framework, compared with deterministic methods. The results show that our generative framework is able to discover a more diverse set of high-quality node mappings.
>
> ---
>
> ### **Time-complexity**
>
> First, we would like to clarify that **the diffusion model does not introduce additional quadratic time complexity** compared to previous frameworks. It only incurs a linear time complexity with respect to the number of reverse steps. The computational cost of each reverse step comes from the forward pass of the network, which has the same time complexity as in prior work.
>
> Regarding the efficiency comparison with MATA*, MATA* is faster than DiffGED **only on small graphs (AIDS, Linux)**, and the inference time of MATA* grows exponentially with graph size and cannot finish within a reasonable time on large graphs (IMDB). In contrast, our DiffGED remains stable and efficient on IMDB dataset. To analysis this, we provide a detailed break down of the inference time:
>
> - Phase 1: DiffGED generates $k$ node matching matrices in parallel through $S=10$ iterative denoising steps (i.e., network forward passes), while MATA* predicts a single node matching matrix in one step. Therefore, MATA* requires less time in node matching matrix prediction.
>
> - Phase 2: DiffGED works by directly extracting $k$ node mappings in parallel from the generated node matching matrices via an efficient greedy approach to derive GED, whereas MATA* approximates GED by applying A* search to the search space pruned by the predicted node matching matrix. For small graphs, both DiffGED's node mappings extraction and MATA*'s A* search are computationally efficient, so MATA* achieves a shorter overall inference time on small graphs due to its faster Phase 1. However, as graph size increases, the computational cost of A* search grows exponentially, dominating MATA*'s total inference time and becoming extremely large on larger graphs (e.g., IMDB). In contrast, the parallel greedy node mapping extraction of DiffGED remains efficient, resulting in a more stable and efficient inference time on larger graphs (IMDB).
>
> Moreover, the running time evalution on larger graphs IMDB further demonstrates that the diffusion process does not introduce quadratic time compelxity.
>
> Notably, the inference time of diffusion model can be further accelerated by reducing the number of denoising steps $S$ via methods such as consistency training [1], but this is not the main focus of our work, we will leave it for future works.
>
>         [1] Song, Y., Dhariwal, P., Chen, M., & Sutskever, I. (2023). Consistency models.
>
> ---
>
> ### **Real-world application**
>
> One of the fundamental applications of GED is graph similarity search in databases. Given a query graph, the goal is to retrieve the top-k most similar graphs or rank all graphs in the database according to GED. We have evaluated this application in our experiments.
>
> Specifically, for all evaluation, each query test graph is paired with $100$ candidate graphs. $\rho$ and $\tau$ measure the matching ratio between the similarity ranking of candidate graphs based on the predicted GEDs and the ground-truth GEDs. p@10/20 further measure the ratio of predicted top-10/20 similar candidate graphs within the ground-truth top-10/20 similar candidate graphs. Results demonstrate that our DiffGED can outperform baselines in this real-world application setting.

---

> > ### Author Rebuttal · Reviewer_z28D · 2026-04-02
> >
> > Thank you to the authors for their response. However, the concerns with the technical claims of this paper have not been addressed. Further clarifications on these concerns below.
> > 1. Thanks to the authors for addressing how the initial question motivation was determining whether the predicted node-matching matrix from DiffGED indicated multimodality. The diversity measurement within Figure 7 certainly shows the capacity of DiffGED and DiffGED-single to produce more distinct edit paths with either predicted or ground-truth GED. Indeed, this serves as a solid ablation to show that the iterative top-k approach integrated into DiffGED-single does not expand the diversity of distinct paths explored. However, the final paragraph in Section 5.2 makes multiple comparative claims to how "the top-k approach used in existing matching-based approaches is limited to generating only a few [distinct edit paths]" (Lines 413-414) and "the iterative approach used in existing frameworks produces highly-correlated node mappings towards one mode, which limits its ability to capture the range of possible edit paths, thus could easily fall into sub-optimal results" (Lines 418-422). This conclusion is drawn from DiffGED's empirical effectiveness over DiffGED-single in Figure 7, but does not provide evidence to eliminate how other aspects of existing matching-based methods, beyond the top-k approach, fails to effectively learn multimodality. Therefore, the lack of a direct comparison to baseline methods (i.e GEDGNN, GEDIOT, etc.) makes the claimed ineffectiveness from Figure 7 to capture distinct edit paths overly-absolute.
> > 2. Thank you to the authors for including details on the variations in efficiency between MATA and DiffGED. To be clear, the initial review question on time complexity of discrete diffusion was not about incurring additional cost relative to GED models from previous work. The initial question was about fundamental limitations of discrete diffusion when translating to large-scale graphs beyond IMDB (a potential research direction). DiffGED certainly benefits from an efficiency boost provided by: parallelization, DDIM sampling for reverse steps at inference, and greedy node-map extraction. However, the DiffMatch formulation (Section 4.2, Algorithms 2 and 3) indicates a (de)noising process consistent with DiGress. This review emphasizes that DiGress, and the current DiffMatch, have quadratic worst-case complexity (see Section F.3 in [1]). This is due to the marginal transition matrix effectively predicting over the entire adjacency structure, which occurs in both training and inference. DiffGED's improved performance and inference speed is sufficient as a contribution for better GED computation. It's also agreed that further efficiency and scalability improvements to DiffGED can be reserved for future work. However, it is critical to be as transparent as possible on framework design decisions and their impacts. The authors mention how denoising during inference "can be computationally expensive" (Line 231), but attribute the expensive computation to how the forward diffusion process is typically 1000 steps. This statement can be misleading, especially without prior knowledge, since the explanation sidesteps why discrete diffusion is expensive. The quadratic time complexity of discrete diffusion is a fundamental limitation and not a shortcoming directly from DiffGED. Based on how DiffMatch is originally presented, the authors response is incorrect (especially when considering the full end-to-end diffusion process) and almost certain to be misleading if included in final publication.
> > 3. Thank you to the authors for describing a potential practical application of DiffGED. Future work applying DiffGED for graph similarity search on databases is likely to produce interesting and helpful insights.
> >
> > [1] Defog: Discrete flow matching for graph generation. ICML 2025.

---

> > > ### Author Response · Authors · 2026-04-03
> > >
> > > ### **R1. Diversity**
> > >
> > > - The reason for using DiffGED-single in the diversity evaluation is to clearly compare the effect on diversity between our generative top-k approach and the deterministic sequential top-k approach used in previous works (here the baseline method is the sequential top-k strategy adopted in prior works GEDGNN/GEDIOT). Since existing methods cannot incorporate our generative top-k approach, we instead integrate the sequential top-k approach into our DiffGED framework to construct DiffGED-single, enabling a fair and controlled comparison.
> > >
> > > - As requested by the reviewer, below we further provide a direct comparison with matching-based baselines GEDGNN and GEDIOT:
> > >   |Dataset|Models|#Paths for GT-GED|#Paths for Pred-GED|
> > >   |-|-|-|-|
> > >   |AIDS|GEDGNN|3.92|9.40|
> > >   ||GEDIOT|3.872|7.876|
> > >   ||DiffGED|11.38|11.62|
> > >   |Linux|GEDGNN|7.243|7.930|
> > >   ||GEDIOT|6.16|7.086|
> > >   ||DiffGED|18.65|18.65|
> > >   |IMDB|GEDGNN|17.153|29.946|
> > >   ||GEDIOT|16.818|28.832|
> > >   ||DiffGED|78.88|79.08|
> > >
> > >   It is clear that DiffGED can produce more number of diverse and high-quality paths, especially in terms of the number of distinct paths corresponding to the ground-truth GED (high-quality paths).
> > >
> > > Notably, A*-search-based baselines (e.g., Noah, GENN-A*, MATA*) do not simply work by producing candidate paths; thus, their diversity is not accessible.
> > >
> > > ---
> > >
> > > ### **R2. Quadratic time complexity**
> > >
> > > Thanks for clarifying your question, we would like to clarify the reviewer’s misunderstanding regarding the quadratic complexity.
> > >
> > > - In our work, the diffusion process is only applied to the node matching matrix (not to the node/adjacency matrix of the graph), it is different from the graph generation.
> > >
> > > - We would like to clarify that **the quadratic complexity is NOT due to the diffusion model**, it is caused by the nature of problem formulation. As stated in Section F.3 of [1], the quadratic worst-case complexity for graph generation comes from complete graph modeling (i.e., adjacency matrix modeling). In our case, we don't model the adj matrix (the diffusion process is not applied to the graph), the quadratic complexity only arises from modeling the node matching matrix of size $|V||V'|$. **However, this is NOT only for the diffusion model; existing frameworks (e.g., GEDGNN, GEDIOT, MATA) also require modeling node matching matrix of size $|V||V'|$ during both training and inference, so they all require a quadratic complexity of $O(|V||V'|)$ as well.** Therefore, the quadratic complexity is a systematic issue inherent to matching-based GED problem formulation rather than a limitation specific to the diffusion model in our method.
> > >
> > > - As stated in [2], the time complexity of matching matrix prediction in GEDGNN can be approximated as $O(max(|V|,|V'|)^2)$ (GEDIOT follows the same complexity). For a single diffusion process in our DiffMatch, the complexity can be approximated as $O(S\cdot max(|V|,|V'|)^2)$. Therefore, the primary difference between DiffMatch and prior deterministic matching-based frameworks (e.g., GEDGNN, GEDIOT) lies in the number of prediction steps $S$.
> > > But indeed, since the diffusion model requires multiple prediction steps $S>1$, it could suffer more from the quadratic complexity compared to baselines due to the repeated quadratic computation.  **This overhead can be reduced by applying methods such as consistency training to decrease $S$ (a mainstream research direction for accelerating diffusion models), or by using latent diffusion to reduce the size of $|V||V'|$ in the intermediate steps of the diffusion process; but this is not the focus of our work, we will mention it in the paper for potential future works.**
> > >
> > > - In Appendix D.3 (Table 6), we further replace the diffusion model with alternative one-step generative models (e.g., VAEs). The results show that the running time of the diffusion model scales approximately linearly with $S$ (i.e., within a factor of $\times S$), and the results do now show any additional quadratic complexity.
> > >
> > > - As stated in Section F.3 of [1], another way to determine whether our diffusion framework introduces any framework-specific quadratic complexity is to evaluate the memory usage of a single diffusion process. As demonstrated in Response 6 to Reviewer ejSp, our diffusion framework does not incur any additional quadratic space complexity compared to baselines (i.e., GEDGNN/GEDIOT).
> > >
> > > - Our framework focuses on generative modeling vs. deterministic modeling; reducing the $|V||V'|$ time complexity that exists in all previous frameworks is not our focus, but can be a potential direction for future GED works, we will mention this in the paper.
> > >
> > > [1] Defog: Discrete flow matching for graph generation. ICML 2025.
> > >
> > > [2] Computing Graph Edit Distance via Neural Graph Matching

---

### Official Review · Reviewer_VNby · 2026-03-09

**Soundness:** 3
**Presentation:** 3
**Significance:** 3
**Originality:** 3
**Overall Recommendation:** 4
**Confidence:** 2

**Summary:**

In this paper, the authors propose DiffGED, a diffusion-based generative method for computing graph edit distance by reframing matching-based graph edit distance computation as conditional generation of node matching matrices. Unlike prior hybrid methods that predict a single deterministic matching matrix and then iteratively extract top-$k$ node mappings, DiffGED uses a diffusion model conditioned on a graph pair to generate $k$ diverse matching matrices in parallel from random initializations, then extracts a top-1 mapping from each matrix to form $k$ candidate edit paths and selects the best one. Experiments are presented on real-world datasets and show that DiffGED achieves accuracy close to exact GED while running faster than existing hybrid approaches.

**Compliance With Llm Reviewing Policy:**

Affirmed.

**Key Questions For Authors:**

Please refer to the Strengths And Weaknesses.

**Limitations:**

Please check the above.

**Strengths And Weaknesses:**

**Strengths**

- The method can generate an arbitrary number $k$ of candidate matching matrices at inference time by sampling $k$ random initial matrices, so it does not need multiple prediction heads or retraining when $k$ changes.

- The proposed two-phase pipeline is clear and easy to follow.

- The paper provides good visual examples that help understanding.

**Weaknesses**

- The sampling cost still scales with both the number of reverse steps $S$ and the number of samples $k$, so large $k$ can be expensive even if some parts are parallelized.

- The Phase 2 greedy extraction is not optimal and can lose quality even if $\hat{M}$ is good.

- The experiments are not evaluated on large graph datasets.

---

> ### Author Rebuttal · Authors · 2026-03-30
>
> ### **W1. Sampling cost**
>
> - Regarding the sampling cost with respect to the number of reverse steps $S$, $S$ can be set to a small fixed number (e.g., $S=10$) during inference. Notably, the the number of reverse steps $S$ during inference can be further reduced via methods such as consistency training [1], but this is not the focus of our work, we will leave it for future works.
>
>         [1] Song, Y., Dhariwal, P., Chen, M., & Sutskever, I. (2023). Consistency models.
>
> - Regarding the sampling cost with respect to the number of samples $k$, previous methods can only sample $k$ candidate solutions sequentially; thus, they are less flexible and cannot leverage parallel computing of GPU to reduce running time. In contrast, our framework enables batched computation of candidate solutions on GPU, it is more flexible: under limited computational resources, the $k$ matching matrices can be computed sequentially, while under abundant computational resources, they can be computed in parallel to reduce the running time. Moroever, as demonstrated in Figure 6, as $k$ increases, the running time of our parallelized method grows slowly compared to previous sequential method.
>
> ---
>
> ### **W2. Greedy extraction**
>
> - In the last part of the ablation study (see Appendix D.3 and Table 7), we evaluate the performance of DiffGED using greedy node mapping extraction vs. exact Hungarian algorithm. The results show that greedy node mapping extraction achieves identical performance to the exact Hungarian method, but with shorter running time.
>
> - We also provide a case study in Figure 9 to compare greedy node mapping extraction with the exact Hungarian method. Since DiffMatch produces high-quality and sparse matching matrices, greedy extraction yields the same node mappings as the exact method, while achieving lower runtime.
>
> ---
>
> ### **W3. Large graph**
>
> We evaluated DiffGED on the IMDB dataset, which is widely used for large-graph evaluation in most of the previous GED works. Due to the NP-hard and cross-graph matching nature of GED, IMDB is considered large for GED. This can be demonstrated in Table 1 that A*-based hybrid methods cannot complete evaluation on IMDB within a reasonable time. In contrast, DiffGED maintains strong performance with stable and efficient running time.
>
> Furthermore, in Appendix D.1, we evaluate the generalization ability to large graphs by training on small graphs and testing on larger graphs from IMDB. The results show that DiffGED generalizes better to larger graphs compared to baseline methods. Moreover, in Appendix D.2, we evaluated DiffGED under unsupervised training setting, it can still perform well without ground-truth supervision on large graph IMDB dataset.
>
> Notably, since GED requires constructing graph pairs (e.g., a dataset of $1000$ graphs can yield $1000^2$ graph pairs) and performing cross-graph node matching, it is infeasible to train and evaluate any existing methods on very large graphs using a single 3090 GPU. Moreover, due to the NP-hard nature of GED (e.g., there are $20! > 2\times10^{18}$ possible node mappings for a grpah pair with only 20 nodes each), no large-scale GED dataset currently exists. Our paper focuses on comparing deterministic vs. generative approaches under the commonly used datasets and evaluation settings, building large-scale benchmarks is not the focus of this work.

---

> > ### Author Rebuttal · Reviewer_VNby · 2026-03-31
> >
> > Thank you for the clarifications. I have no additional questions.

---

### Official Review · Reviewer_BWas · 2026-03-10

**Soundness:** 2
**Presentation:** 3
**Significance:** 3
**Originality:** 3
**Overall Recommendation:** 4
**Confidence:** 4

**Summary:**

This paper proposes a denoisiong-diffusion-based model for predicting solutions (rather than just regressing the value) to the graph edit distance (GED) problem, which can be cast as a graph matching problem. The prediction of soft graph matchings (i.e. a continuous relaxation) is indeed formulated as a generative task, and the soft matchings are then rounded to 0-1 matchings via a greedy and fast assignment algorithm. Interestingly, diffusion produces several diverse soft matchings, and the best hard matching can be extracted out of the ensemble of soft ones. The method is benchmarked on the classical Bai et al. datasets, and shows very promising solution quality and efficient inference times.

**Compliance With Llm Reviewing Policy:**

Affirmed.

**Final Justification:**

In light of the new results and inclusion of one new baseline, I will raise my score. However, I do so in good faith that the authors will include a honest discussion about dataset splitting and the associated performance deterioration, and about the reasons for excluding some baselines (e.g. they only provide regression values, no public code, no access to Gurobi).

**Key Questions For Authors:**

See weaknesses.

**Limitations:**

I would suggest the authors to discuss the limitations of the approach and possible directions for improvement.

**Strengths And Weaknesses:**

Strengths:
- the use of diffusion is an interesting new perspective on the GED prediction task. I think that it is very promising, as it likely breaks graph symmetries and could mimic the reversal of a local search algorithm (such as Refine).
- Related to the point above, the use of diffusion lends itself to ensembling and therefore to exploring the search space of matchings more broadly.
- the paper is well-written, it is intuitive yet rigorous at the same time.
- The results encompass several metrics, including inference time. This makes it easy to understand the speed/accuracy trade offs of the various models. Towards this end, the ablation in Table 7 is very interesting.

Weaknesses:
- The paper states “matching-based methods formulate GED computation as a bipartite graph matching problem and can be solved in polynomial time.” This is false, as formulating GED as a bipartite graph matching problem while keeping the quadratic terms (i.e., a quadratic assignment problem) is still NP-hard. These heuristics drop the quadratic terms. You should rephrase it as “linear-assignment-based heuristics” or similar.
- The paper also states “However, this approach is deterministic”. Being deterministic is a positive aspect. I understand the argument on the local sub-optimal minima, but determinism is not the correct way to argue it, in my opinion.
- The paper assumes unit edit costs, no edge labels, and symmetry (GED(G, G′ ) = GED(G′ , G)). These are restrictive assumptions that could limit the applicability of the method. For example, under these assumptions it is impossible to reduce subgraph isomorphism to GED.
- The paper reports in the main body the results on the datasets from Bai et al., that have *data leakage* between training and test set (e.g., see Roy et al. ICML 2025. Also, you are missing the year in that citation.), while it hides the results on the de-leaked datasets in the appendix (Table 4). Here, we can clearly see a drop in the predictive performance.
- Also, it should be clearer whether the test pairs have both pairs belonging to the test split, or if one belongs to training. My understanding is that Table 1 had data where one graph is from the *training* set, while Table 2 in the appendix only contains test pairs. Again, this can be seen as some sort of leakage, and Table 2 shows indeed worse results.
- The paper only uses thee datasets. More recent papers like [3], [4] use more datasets.
- The paper does not compare to strong baseline methods. Among non-neural methods, ILP formulations like F2 are usually very competitive. Most importantly, it does not compare to recent methods like GRAPHEDX [1], App-BMao [2] and most importantly Grail [3], whose generated codes provide fast and accurate heuristics, likely better than DiffGed. In fact, Gelato [4] yields even better solution quality. The authors should compare at least to Grail, as it was published one year ago and seems to show better performance than DiffGed.
- It is unclear how DiffGed will generalize to larger graph sizes, which would be the main application of GED prediction models.

Overall, I think that the paper has some potential, but the experimental evaluation should be more thorough, especially with respect to baselines and with respect to data leakage.

.

- [1] Roy et al. Graph Edit Distance with General Costs Using Neural Set Divergence. LoG 2024.
- [2] Mouyi Xu and Lijun Chang. Graph Edit Distance Estimation: A New Heuristic and A Holistic Evaluation of Learning-based Methods. Proc. ACM Manag. Data 2025.
- [3] Verma et al. GRAIL: Graph Edit Distance and Node Alignment using LLM-Generated Code. ICML 2025.
- [4] Pellizzoni et al. Gelato: Graph Edit Distance via Autoregressive Neural Combinatorial Optimization. ICLR 2026.

---

> ### Author Rebuttal · Authors · 2026-03-30
>
> ### **W1. Deterministic**
>
> Deterministic is not a drawback for standard prediction tasks. But for GED, the solution is often selected from a set of candidate solutions in the search space, deterministic methods are limited in exploring diverse candidates. We provided ablation studies to demonstrate that our generative formulation significantly improves the performance.
>
> ---
>
> ### **W2. Problem definition**
>
> - We followed the standard GED definition, datasets, and evaluation settings widely used in most of prior works.
>
> - Our focus is deterministic vs. generative modeling rather than complex matching mechanisms. Variable edit costs/edge labels/symmetry does not affect our generative formulation, they only affect the ground-truth labels. Edge labels can be easily incorporated as input edge features in GNNs. **Future works that focus on other settings or matching mechanism can also adopt our generative formulation**. Thus, using the standard GED definition does not limit the applicability of our method.
>
> - In the response to W5, we include a dataset with edge labels (ZINC).
>
> ---
>
>
> ### **W3. Leakage**
>
> We want to clarify that the "leakage" only refers to the existence of isomorphic graphs (i.e., GED=0) in the dataset.
>
> The results on the de-leaked datasets (Table 4) **do not show a significant performance drop** compared to the results in Tables 1&2, especially when compared to baselines:
>
> - AIDS: The accuracy of DiffGED drops by only around 2% (i.e., from 98% to 96% and from 96.4% to 96%), which is very small, and DiffGED still yields a very large performance gap compared to baselines.
>
> - Linux: The accuracy of DiffGED drops from 100% to 92.4% and 86.1%, while the accuracy of baselines drops from around 95% to around 65% and 55%.
>
> - IMDB: The accuracy of DiffGED drops by less than 1%, while the accuracy of baseline methods drops by around 10%.
>
> Compared to the performance drop of baseline methods, the performance drop of DiffGED is very small, and DiffGED still yields a large performance gap.
>
> ---
>
> ### **W4. Test pairs**
>
> - We followed prior works to use both train-test pairs and test-test pairs for evaluation. The train-test pair setting is commonly used for graph similarity search evaluation (i.e., ranking-based metrics), where the training graphs are used as candidate graphs in the database and the testing graphs are used as query graphs.
>
> - Comparing Tables 1 and 2, there is almost **no performance drop** for DiffGED. For AIDS, the accuracy only drops from 98% to 96.4%, which is insignificant, and the accuracy on Linux and IMDB does not drop at all.
>
> ---
>
> ### **W5. Datasets & Baselines**
>
> - Datasets: We used the most common open-sourced GED datasets with ground-truth matchings provided. [3] didn't provide the ground-truth matchings. [4] was just accepted by ICLR26, very close to the ICML deadline. Below we provide results on two extra datasets ZINC (edge-labeled) and Code2.
>
> - Baselines: F2 is often used for ground-truth computation but is very slow, and we currently do not have access to it (requires Gurobi). GRAPHEDX is a regression-based method, while our evaluation is based on feasible discrete GED derived from node matching. Grail and Gelato didn't follow the commonly used metrices/dataset construction/GPU adopted in our work and prior works (e.g., their testset construction didn't consider graph similarity search), thus their results are not directly comparable just by inspection. Below we provide the performance of Grail as requested.
>
> |Datasets|Models|MAE|Accuracy|Time|
> |-|-|-|-|-|
> |ZINC|GEDGNN|5.198|6.6%|0.701|
> ||GEDIOT|5.882|4.1%|0.717|
> ||Grail|1.777|20.4%|0.012|
> ||DiffGED|0.387|70.8%|0.129|
> |Code2|GEDGNN|1.237|62.6%|1.051|
> ||GEDIOT|2.588|36.1%|1.055|
> ||Grail|0.635|63.9%|0.045|
> ||DiffGED|0.04|97%|0.219|
> |AIDS|Grail|0.204|83.7%|0.005|
> ||DiffGED|0.024|96.4%|0.075|
> |Linux|Grail|0.01|99.5%|0.00333|
> ||DiffGED|0.0|100%|0.069|
> |IMDB|Grail|0.025|99.3%|0.21144|
> ||DiffGED|0.932|94.6%|0.151|
>
> Due to the character limits, we only report the MAE and Accuracy.
>
> On additional datasets ZINC and Code2, DiffGED still outperforms all baselines.
>
> Compared with Grail, DiffGED achieves better MAE and Accuracy on ZINC, Code2, AIDS, and Linux. Notably, the ground-truth GEDs for IMDB evaluation are approximated since it contains large graphs, the results on IMDB only indicate that Grail is closer to the approximated ground-truth, rather than the actual ground-truth (approximated ground-truth ≥ actual ground-truth).
>
> Moreover, the heuristics generated by Grail are faster on small graphs, but their running time increases significantly with graph size and becomes slower than DiffGED on large graphs (e.g., IMDB). In contrast, DiffGED has a more stable running time.
>
> Also, Grail uses LLMs heavily to generate heuristics, which is very costly. In contrast, DiffGED is trained only using a single 3090 GPU, which is much cheaper.
>
> ---
>
> ### **W6. Large graphs**
>
> Please refer to Appendix D.1 and the response W3 to reviewer VNby.

---

> > ### Author Rebuttal · Reviewer_BWas · 2026-04-03
> >
> > > In the response to W5, we include a dataset with edge labels (ZINC)
> >
> > Great, I think this will strengthen the paper.
> >
> > > The results on the de-leaked datasets (Table 4) do not show a significant performance drop
> >
> > This is false. A drop from 98% to 96% means that the error rate **doubles**. This is confirmed by the results on the MAE, which increases from 0.022 to 0.046. For the intra-test error, the error rate increases by **4 times**.
> > The results are fine, but they should be presented honestly to the reader.
> >
> > > Datasets: We used the most common open-sourced GED datasets with ground-truth matchings provided. Below we provide results on two extra datasets ZINC (edge-labeled) and Code2. ... Below we provide the performance of Grail as requested.
> >
> > I think this will also strengthen the paper.
> >
> > ---
> > In light of the new results and inclusion of one new baseline, I will raise my score. However, I do so in good faith that the authors will include a honest discussion about dataset splitting and the associated performance deterioration, and about the reasons for excluding some baselines (e.g. they only provide regression values, no public code, no access to Gurobi).

---

> > > ### Author Response · Authors · 2026-04-03
> > >
> > > Thanks for raising the score and for clarifying your perspective on performance change.
> > >
> > > We agree that, when viewed in terms of relative performance change (%), the MAE/error rate appears to double. But we would also like to mention that the magnitude of original MAE (and error rate) of DiffGED is already very small. In such a low-error regime, even a minor absolute increase can result in a large relative change. In contrast, baseline methods start from a much higher error level, where even substantial absolute degradation would not translate into a similarly large relative change. Thus, we primarily focus on absolute performance changes and did not initially interpret the reviewer’s comment from the perspective of relative change. Again, thanks for clarifying your perspective.
> > > As suggested by the reviewer, we will mention the relative performance changes in the paper, including the doubled error rate.
> > >
> > > We would also like to clarify that the results for test–test pairs and the de-leaked setting are included in the appendix solely due to the 8-page limit. And we followed common practice in many prior works to present the train–test results (which considers the context of graph similarity search) as the primary table in the main paper (this was initially used in [1], and most later works have followed this setting) .
> > >
> > >     [1] https://pytorch-geometric.readthedocs.io/en/latest/generated/torch_geometric.datasets.GEDDataset.html
> > >
> > > We would also like to mention that most prior works (e.g., GEDGNN, GEDIOT, and Grail) did not report results on de-leaked datasets, whereas we explicitly include them in our work. Therefore, we are presenting all the results honestly, and their placement in the appendix is solely due to the space limitation of the main paper.
> > > In the camera-ready version, which allows an additional page, we will move the test–test and de-leaked results to the main paper and include the corresponding discussion as suggested by the reviewer.

---

### Official Review · Reviewer_ejSp · 2026-03-13

**Soundness:** 2
**Presentation:** 3
**Significance:** 3
**Originality:** 2
**Overall Recommendation:** 4
**Confidence:** 3

**Summary:**

The paper introduces DiffGED, a diffusion-based framework that formulates graph matching as a generative task. Instead of predicting a single node matching matrix, the model generates multiple diverse node matching matrices using a diffusion model (DiffMatch). Candidate node mappings are then extracted from these matrices to produce multiple edit paths, from which the best GED estimate is selected. Experiments on several benchmark datasets show that DiffGED achieves strong performance in terms of accuracy and runtime compared with existing hybrid approaches.

**Compliance With Llm Reviewing Policy:**

Affirmed.

**Key Questions For Authors:**

1. Since training requires ground truth node mappings or GED values, how scalable is the dataset generation process for larger graphs?

2. How much does the greedy node mapping extraction affect final GED accuracy compared to using an exact assignment algorithm such as the Hungarian method?

3. How well does DiffGED generalize to graphs that are significantly larger or structurally different from those in the training set?

4. Could the authors provide ablations showing the contribution of the diffusion component compared with simpler stochastic sampling approaches?

5. What are the memory and computational requirements of DiffMatch when generating many matching matrices (large k)?

**Limitations:**

The authors have not discussed the limitations and potential negative societal impact of their work.

The authors could discuss the dependence on ground truth GED or node mappings during training, which may require expensive exact GED computation and limit scalability to larger graphs. In addition, the method relies on diffusion-based generation and multiple matching matrices, which may introduce significant computational and memory overhead for very large graphs or large values of k.

**Strengths And Weaknesses:**

1. Soundness:

The paper proposes an interesting idea of using diffusion models to generate multiple node matching matrices for GED computation. The formulation is reasonable and the algorithmic pipeline is clearly defined.

However, some aspects of the method is not very sound. First, the approach relies on ground truth node mappings or edit paths during training, which may require expensive exact GED computations. Since GED itself is NP-hard, generating such supervision could become a bottleneck when scaling to larger graphs.

Second, the extraction of node mappings uses a greedy algorithm rather than an optimal solver, which does not guarantee optimal node mappings even if the predicted matching matrix is accurate. The paper acknowledges this limitation but does not thoroughly analyze how often this approximation affects final GED results.

Also while the experimental results are strong, the reported improvements (likw near-perfect accuracy in some datasets) appear unusually large compared with prior work, which raises questions about potential dataset biases or evaluation settings. Additional ablation studies or robustness analysis would strengthen the technical claims.

2. Presentation:

The paper is generally well structured and easy to follow. The motivation for introducing generative diversity into graph matching is clearly explained, and the overall pipeline of DiffGED is presented with helpful figures and diagrams.
However, some sections, like the diffusion model formulation and the denoising network architecture are quite dense and may be difficult for readers unfamiliar with diffusion models. More intuitive explanations or simplified summaries of the design choices could improve readability.

3. Significance:

Graph edit distance remains an important problem in many domains. Improving the scalability and quality of approximate GED computation is therefore a meaningful research direction.

If the proposed generative approach consistently produces diverse and accurate edit paths, it could be useful for applications where multiple candidate graph alignments are beneficial. However, the practical significance may depend on how well the method scales to larger graphs and more diverse datasets.

4. Originality:

The paper introduces a novel generative formulation of graph matching using diffusion models, which appears to be relatively unexplored in the GED literature. While the underlying components, diffusion models, GNNs, and graph matching are well established, the idea of generating multiple diverse matching matrices through diffusion represents a creative combination of existing techniques.

---

> ### Author Rebuttal · Authors · 2026-03-30
>
> ### **1. Ground-truth for training**
>
> We would like to clarify that all previous methods require ground-truth training labels. Our work focus on comparing deterministic vs. generative methods under the same evaluation setting commonly adopted in prior works, and unsupervised learning is not the focus of our work.
>
> Regarding training label generation/unsupervised training for larger graphs, we adopted two strategies:
>
> - The training labels for larger graphs (e.g., IMDB) are generated synthetically by applying random edit operations to real graphs (see Appendix C.1);
>
> - In Appendix D.2 (Table 5), we trained our framework and the baselines using the most recent unsupervised training method GEDRanker, which does not require any ground-truth labels.
>
> Under both strategies, our method can significantly outperform previous methods.
>
> ---
>
> ### **2. Greedy node mapping extraction**
>
> - In the last part of Appendix D.3 (Table 7), we evaluate the performance of DiffGED using greedy node mapping extraction vs. exact Hungarian algorithm. The results show that greedy node mapping extraction achieves identical performance to the exact Hungarian method, but with shorter running time.
> - We also provide a case study in Figure 9 to compare greedy node mapping extraction with the exact Hungarian method. Since DiffMatch produces high-quality and sparse matching matrices, greedy extraction yields the same node mappings as the exact method, while achieving lower runtime.
>
>
> ---
>
> ### **3. Generalization ability**
>
> In Appendix D.1 (Tables 2, 3 & 4), we performed several generalization evaluation on unseen graphs / datasets without isomorphic graphs / larger graphs. The results demonstrate that DiffGED outperforms baseline methods under all generalization evaluation settings.
>
> ---
>
> ### **4. Near-perfect accruacy of DiffGED**
>
> - We strictly followed the same evaluation settings and datasets as those commonly adopted in prior works. We also provided the source code with unified implementations of all baselines, which ensures a fair and consistent comparison.
>
> - For the potential dataset biases, in Appendix D.1 (Table 4), we conducted evaluation by removing the isomorphic graph pairs to address the potential dataset biases claimed in [1]. The results show that DiffGED can still yield a significant performance gap compared to previous methods, and the performance drop of DiffGED is small compared to the performance drop of baseline methods, thus our framework is robust.
>
>         [1] Position: Graph matching systems deserve better benchmarks.
>
> - For the near-perfect accuracy of DiffGED, we have provided detailed ablation studies in Appendix D.3 to analyze such excellent performance. The near-perfect accuracy across all datasets mainly comes from our novel generative formulation and the iterative generation capability of the diffusion model.
>
> ---
>
> ### **5. Ablation study on diffusion component**
>
> In Appendix D.3 (Table 6), we conduct detailed ablation studies on the diffusion component, comparing it with simpler stochastic sampling approaches:
>
> - In "Do we really need generative modeling?", we removed our proposed generative modeling and simply inject random noise into the deterministic matching matrix predicted by previous methods to perform stochastic sampling (see GEDGNN (Noise)). The results in Table 6 show that simpler stochastic sampling approach does not work and may hinder the performance of previous methods. This demonstrates the importance of our generative formulation.
>
> - In "Do we really need diffusion model?", we replaced the diffusion model with other generative models (see DiffGED (w/o diffusion) and DiffGED (VAE)). The results demonstrate the effectiveness of using the diffusion model.
>
> ---
>
> ### **6. Computational requirement**
>
> - The diffusion process does not introduce any extra memory overhead; the memory requirement for a single diffusion process (i.e., $k=1$) is almost the same as prior deterministic methods. Below are the max memory requirements of DiffGED and baselines for $k=1$ on a single graph pair:
>
>   |Dataset|Models|GPU memory (MB)|
>   |-|-|-|
>   |AIDS|GEDGNN|1294|
>   ||GEDIOT|1296|
>   ||DiffGED|1222|
>   |Linux|GEDGNN|1294|
>   ||GEDIOT|1296|
>   ||DiffGED|1222|
>   |IMDB|GEDGNN|1296|
>   ||GEDIOT|1298|
>   ||DiffGED|1230|
>
> - The memory overhead mainly comes from the parallelized computation of $k$ matching matrices; thus, the memory overhead increases linearly with $k$. But this is not a drawback, since our framework enables batched computation of $k$ candidate solutions on GPU, it is more flexible: under limited computational resources, the $k$ matching matrices can be computed sequentially, while under abundant computational resources, they can be computed in parallel to reduce the running time. In contrast, previous methods can only compute $k$ candidate solutions sequentially; thus, they are less flexible and cannot leverage parallel computing of GPU to reduce running time under abundant computational resources.

---

> > ### Author Rebuttal · Reviewer_ejSp · 2026-04-04
> >
> > Thanks for answering my questions. I intend to keep my original score.

---

### Decision · Program_Chairs · 2026-04-30

**Decision:**

Accept (regular)

**Comment:**

This paper proposes DiffGED, a generative diffusion-based framework for graph edit distance computation, reframing graph matching as a generative task to produce diverse matching matrices and candidate edit paths. The reviewers generally agreed that the problem setting is novel and that the idea is technically interesting, with strong empirical results and a clear presentation. The main concerns were about experimental breadth, baseline coverage, and some technical details such as supervision and greedy extraction, but the authors’ rebuttal addressed many of these points by adding ablations, clarifying the evaluation protocol, and reporting additional results. Overall, the discussion supports the view that the paper offers a novel and promising direction for GED computation with meaningful practical value. I recommend (weak) accept.